# Phosphorylation of the F-BAR protein Hof1 drives septin ring splitting in budding yeast

Maritzaida Varela Salgado [1], Ingrid E. Adriaans[1], Sandra A. Touati[2], Sandy Ibanes[1], Joséphine Lai-Kee-Him[3], Aurélie Ancelin[3], Luca Cipelletti [4,5], Laura Picas [6] & Simonetta Piatti [1] ✉

A double septin ring accompanies cytokinesis in yeasts and mammalian cells. In budding yeast, reorganisation of the septin collar at the bud neck into a dynamic double ring is essential for actomyosin ring constriction and cytokinesis. Septin reorganisation requires the Mitotic Exit Network (MEN), a kinase cascade essential for cytokinesis. However, the effectors of MEN in this process are unknown. Here we identify the F-BAR protein Hof1 as a critical target of MEN in septin remodelling. Phospho-mimicking *HOF1* mutant alleles overcome the inability of MEN mutants to undergo septin reorganisation by decreasing Hof1 binding to septins and facilitating its translocation to the actomyosin ring. Hof1-mediated septin rearrangement requires its F-BAR domain, suggesting that it may involve a local membrane remodelling that leads to septin reorganisation. In vitro Hof1 can induce the formation of intertwined septin bundles, while a phosphomimetic Hof1 protein has impaired septin-bundling activity. Altogether, our data indicate that Hof1 modulates septin architecture in distinct ways depending on its phosphorylation status.

Cytokinesis is the ultimate step of cell division, leading to the physical splitting of the two daughter cells' cytoplasm. It is a very complex process, involving a plethora of proteins and displaying a high degree of plasticity. Importantly, cytokinesis failure leads to whole genome duplication, which can drive genetic instability and tumorigenesis[1,2].

Although different modes of cytokinesis can be found across species[3], a contractile ring made of actin filaments and myosin II motors (referred to as actomyosin ring or AMR thereafter) is a common cytokinetic element in amoebas, fungi and animal cells[4]. The AMR is associated with the equatorial plasma membrane and generates the mechanical force for cleavage furrow ingression during cytokinesis[4]. Microtubules, in particular at the spindle midzone, specify cleavage plane positioning and in several organisms are required for furrow ingression and to direct membrane vesicles to the site of cytokinesis[5].

Besides actin and microtubules, septins constitute another class of cytoskeletal proteins that contribute to cytokinesis in many systems. Initially discovered in budding yeast for their essential role in cell division[6], septins were later found in all animal cells, fungi, ciliates and algae, while they are absent in land plants[7,8]. Although they often localise at the division site, their requirement for cytokinesis is variable, depending on the organism and cell type[9]. Septins interact directly with membranes through their high affinity for negatively charged phospholipids, with a preference for phosphatidylinositol 4,5-bisphosphate (hereafter referred to as PI(4,5)P$_2$)[10–14]. Septin binding to membranes is also influenced by membrane curvature, with a marked propensity for micrometric curvatures[15–17]. Accordingly, septins are often found at curved membranes, such as the hyphal branch

[1]CRBM (Centre de Recherche en Biologie cellulaire de Montpellier), University of Montpellier, CNRS UMR 5237, 34293 Montpellier, France. [2]Université Paris Cité, CNRS, Institut Jacques Monod, 75013 Paris, France. [3]CBS (Centre de Biologie Structurale), University of Montpellier, CNRS UMR 5048, INSERM U 1054, 34090 Montpellier, France. [4]L2C (Laboratoire Charles Coulomb), University of Montpellier, CNRS 34095, Montpellier, France. [5]IUF (Institut Universitaire de France, 75231 Paris, France. [6]IRIM (Institut de Recherche en Infectiologie de Montpellier), University of Montpellier, CNRS UMR 9004, 34293 Montpellier, France. ✉e-mail: simonetta.piatti@crbm.cnrs.fr

sites of filamentous fungi, the base of primary cilia and the annulus of sperm tails[15,18–22].

In the budding yeast *S. cerevisiae* septins localise at the bud neck, i.e. the division site, throughout most of the cell cycle. At this location, they recruit most cytokinetic proteins at various moments of the cell cycle and prompt actomyosin ring assembly, hence explaining their strict requirement for cytokinesis[23–25]. This scaffolding function is performed by a rigid collar that is made of axial septin filaments, running along the mother-bud axis, and circumferential septin filaments, perpendicular to the axial filaments[26,27].

At the onset of cytokinesis, the septin collar is dramatically rearranged into two parallel rings, one at each side of the bud neck, that sandwich the constricting AMR. This process, which hereafter we refer to as septin ring "splitting", is an essential prerequisite for actomyosin ring constriction and cytokinesis[28]. Since the septin double ring is made exclusively by circumferential filaments, septin ring splitting has been proposed to be driven by the selective depolymerisation of axial septin filaments, possibly accompanied by septin filament repolymerisation in the orthogonal orientation[27,29].

Septin ring splitting is triggered by the Mitotic Exit Network (MEN)[28], a Hippo-like kinase cascade that ultimately promotes the release from the nucleolus and activation of the protein phosphatase Cdc14[30]. In agreement with its key role in septin ring splitting, MEN is essential for cytokinesis independently of its well-characterised function in promoting mitotic exit[28,31–33]. While several cytokinetic targets of MEN have been identified[34–40], its relevant substrates for septin ring splitting are not known.

The Hof1 protein is involved in cytokinesis and is an established target of MEN[37,41–43]. Hof1 carries at the N-terminus an F-BAR domain, which is thought to sense and induce membrane curvature[44,45], an intrinsically disordered region in the middle that is highly phosphorylated[37] and an SH3 domain at the C-terminus. Its counterpart in *S. pombe*, called Cdc15, is essential for cytokinesis, partly by promoting AMR integrity[46–50]. Additionally, the Hof1/Cdc15 mammalian ortholog PSTPIP1 (Proline-serine-threonine phosphatase-interacting protein 1) localises at the cleavage furrow, suggesting conserved cytokinetic function(s)[51].

Both *S.c.* Hof1 and *S.p.* Cdc15 play a major role in the control of actin polymerisation and bundling, partly by modulating formin activity[46–48,52–54]. Consistently, *S.c.* Hof1 and *S.p.* Cdc15 have been implicated in polarised growth[52,55]. However, the cytokinetic function of Hof1 has been mainly linked to the activation of the chitin synthase Chs2, which associates and constricts with the AMR to catalyse the assembly of the primary septum at the division site. Indeed, *CHS2* overexpression or gain of function mutations can suppress the lethality of *hof1Δ* mutants at high temperatures[43,56].

Hof1 associates with septins in mitosis, forming two narrow-spaced rings at the opposed edges of the septin collar. Around the time of septin ring splitting and cytokinesis, Hof1 leaves the septin collar and joins the AMR, where it partially constricts along with it[37,41,42,57–59]. The sudden translocation from septins to the AMR is triggered by Hof1 phosphorylation catalysed at least partly by the MEN kinase Dbf2, which disengages Hof1 from septins and makes it free to join the AMR[37,42,43]. The striking temporal coincidence between septin splitting and Hof1 arrival at the AMR raises the possibility that Hof1 could play a role in septin reorganisation at cytokinesis. Here, we show that Hof1 contributes to septin ring splitting upon its MEN-dependent relocalisation to the AMR. The role of Hof1 in septin remodelling requires its F-BAR domain, suggesting that it involves a local reorganisation of the plasma membrane at the division site. On the other hand, our in vitro reconstitution assays indicate that Hof1 is able to bundle septin filaments, suggesting that in mitosis it may participate in septin collar architecture and stability.

## Results
### Hof1 contributes to septin reorganisation at cytokinesis
Our previous data highlighted the Mitotic Exit Network (MEN), and in particular its downstream phosphatase, Cdc14, as a crucial trigger of septin ring splitting, raising the possibility that critical dephosphorylations accompany this important transition[28]. However, the MEN kinases Cdc15 and Mob1-Dbf2 are well-characterised targets of Cdc14 at mitotic exit[60,61] and may therefore contribute to septin ring splitting downstream to Cdc14 activation (Fig. 1A). To address if MEN kinases are relevant targets of Cdc14 in septin ring splitting, we introduced the non-phosphorylatable *CDC15-7A*[60] and/or *MOB1-2A*[61] alleles in *GAL1-DMA2* cells, which fail septin ring splitting in galactose-containing medium (*GAL1* promoter on) because of Dma2-dependent MEN inhibition[28,62,63]. *CDC15-7A* significantly improved the ability of *GAL1-DMA2* cells to undergo septin rearrangement at cytokinesis, while *MOB1-2A* had a negligible effect (Fig. 1B). However, the presence of both alleles had an additive effect, with 59% of cells undergoing septin clearance from the bud neck compared to only 23% in the control (Fig. 1B). We conclude that the Cdc15 and Mob1-Dbf2 kinases are important targets of Cdc14 in septin reorganisation at cytokinesis, suggesting that phosphorylation events may be instrumental to this process.

Hof1 is an established phosphorylation target of the Mob1-Dbf2 kinase, which promotes its translocation from the septin collar to the AMR at mitotic exit[37,41,42,57–59]. We carefully examined by time-lapse imaging the precise timing of Hof1 reorganisation from a double ring (marking its association with septins) into a single medial ring (marking its relocalisation to the AMR) relative to septin ring splitting. We found that the double ring of Hof1-eGFP underwent a visible rearrangement on average 2.5 min before septin ring splitting and completely collapsed into a single contractile ring 0.3 min prior to septin ring splitting (n = 30, Fig. 1C), suggesting that Hof1 reorganisation shortly precedes septin remodelling. This prompted us to assess its possible contribution to septin ring splitting. Deletion of budding yeast *HOF1* was reported to cause severe growth and cytokinesis defects, especially at high temperatures (e.g. 37 °C)[41,43,64]. In order to use *hof1* mutants for live cell imaging, we first carefully analysed their phenotypes under different growth conditions. When we deleted one copy of *HOF1* in the diploid strain background that we routinely use (W303), we found that after sporulation and tetrad dissection 100% of the haploid *hof1Δ* meiotic segregants were lethal already at 25 °C in rich medium (YEPD), while a fraction of them could survive at temperatures below 25 °C (Fig. S1A). Surprisingly, the viability of *hof1Δ* haploid spores was significantly improved in synthetic medium (SD), with most spores being able to divide at 25 °C but not at higher temperatures (30 °C and above, Fig. S1A). In all cases, however, *hof1Δ* spores formed colonies that were smaller in size than their *HOF1* counterparts, confirming that *HOF1* is important for the fitness of yeast cells at low temperatures and becomes essential for viability at high temperatures, in line with previous data[41,43]. Growth assays of viable *hof1Δ* meiotic segregants confirmed that they proliferate extremely slowly on YEPD plates at 25 °C and are temperature-sensitive at temperatures above 30 °C, while their fitness was substantially improved in SD medium (Fig. S1B). In agreement with published data[41], *hof1Δ* cells sporadically gave rise to relatively healthy colonies at intermediate temperatures (e.g. 30 °C, Fig. S1B), suggesting that they are prone to adapt or accumulate suppressors with time. Altogether, these data indicate that the phenotype of *hof1Δ* cells is largely influenced by the growth conditions.

We then asked if Hof1 could participate to septin ring splitting. While *hof1Δ* cells displayed no obvious defect in septin splitting under our conventional microscopy conditions (SD medium, 30 °C) consistent with previous observations[58], their incubation for 90 min in YEPD at 30 °C caused the accumulation of 22% of chained cells with

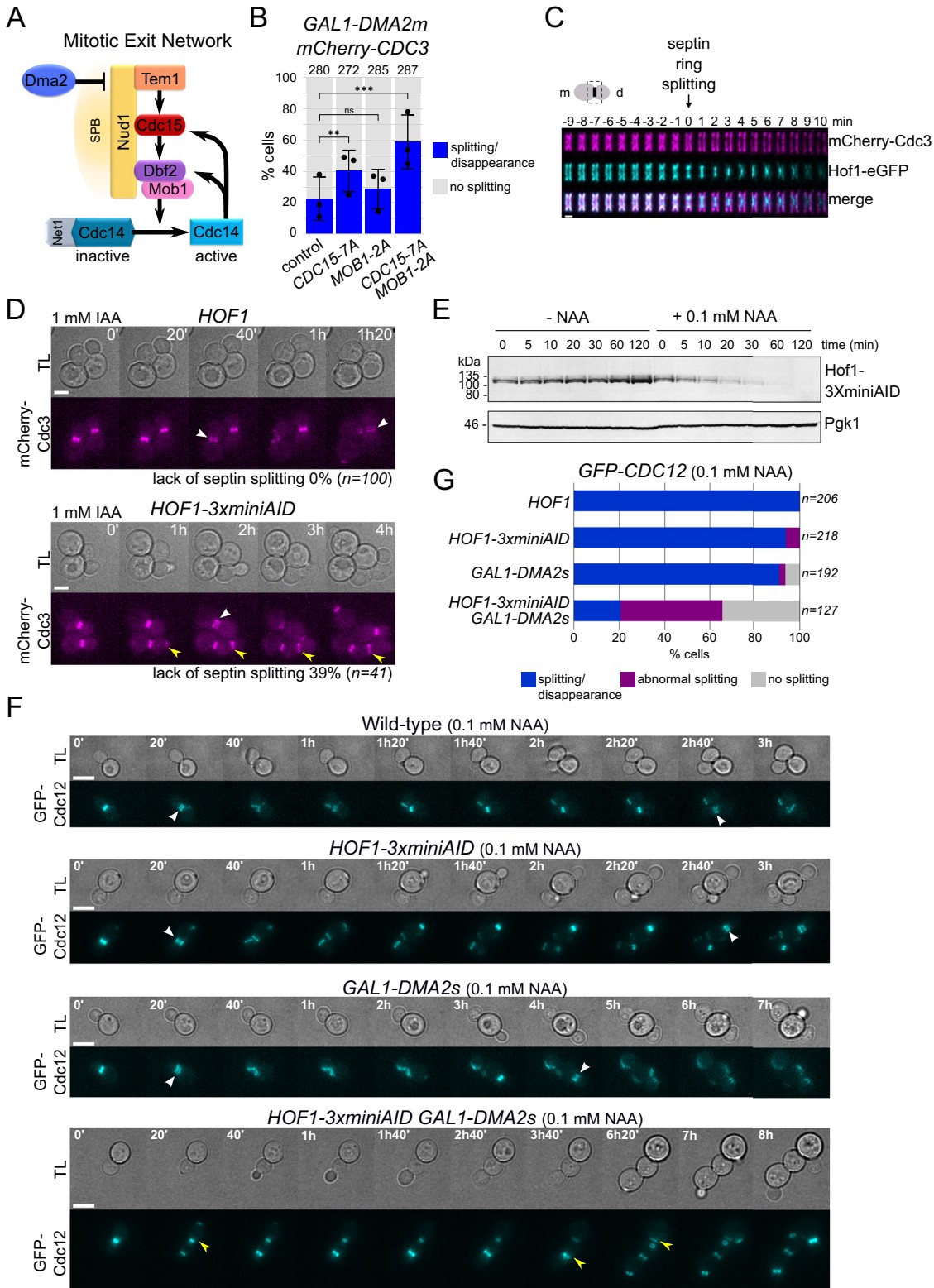

multiple septin collars (Fig. S1C). However, many cells were extremely sick or lysed during treatment, thereby precluding a precise estimation of the septin splitting defects of *hof1Δ* cells. To overcome this problem and avoid the gain of suppressors, we generated a conditional *HOF1* depletion strain by introducing three miniAID tags (AID: auxin-inducible degron[65]) at the 3′ end of the *HOF1* ORF, to allow for the depletion of Hof1 upon expression of the E3 ligase OsTir1 in the presence of auxin-like hormones. Live cell imaging of *HOF1-3xminiAID* cells in the presence of 1 mM of the synthetic auxin indole-3-acetic acid (IAA)

showed that septin ring splitting failed in 39% of the cells (Fig. 1D), indicating that Hof1 is required for efficient septin ring splitting. However, under these conditions cell division was slow even for wild-type cells, and the majority of *HOF1-3xminiAID* cells either lysed or did not attempt to divide. The reason for this might reside in the photo-toxicity caused by an IAA derivative that is generated upon excitation of commonly used fluorophores[66]. In contrast, illumination in the presence of 0.1 mM 1-naphtaleneacetic acid (NAA) is better tolerated by yeast cells[66]. The addition of 0.1 mM NAA to *HOF1-3xminiAID* cells

**Fig. 1 | MEN kinases and Hof1 are involved in septin ring splitting. A** Schematic representation of the Mitotic Exit Network and the proteins relevant for its regulation. SPB: spindle pole body. Dma2 downregulates MEN activity by ubiquitylation of the MEN scaffold at SPBs Nud1, thereby interfering with the recruitment of MEN components[28]. **B** *mCherry-CDC3 GAL1-DMA2m* cells carrying the indicated phosphomimetic mutant alleles of MEN kinases were grown in SD-raffinose (*GAL1* promoter off) and induced for 90 min with galactose before imaging in SD-raffinose/galactose (*GAL1* promoter on) at 30 °C. Septin splitting and lack of splitting events have been quantified in cells undergoing mitotic exit and a new round of budding. Columns represent mean values and error bars SD. The number of cells that were scored are indicated at the top of each column (number of biological replicates $N = 3$). *P*-values were calculated with a two-sided $\chi^2$ test. **\*\***$p = 0.009$; ns not significant ($p = 0.333$); **\*\*\***$p = 2.2681E^{-7}$. **C** Diploid cells expressing mCherry-Cdc3 and Hof1-eGFP were arrested in mitosis by nocodazole treatment and then spotted on agarose pads of SD medium for imaging every 1′ at 30 °C. Images were deconvoluted to better visualise Hof1-eGFP and septin rearrangements and cropped around the bud neck. The time of septin ring splitting has been used as reference (time = 0). **D** *HOF1* and *HOF1-3XminiAID* cells expressing mCherry-Cdc3 were grown in SD at 30 °C and supplemented with 1 mM IAA (3-indolacetic acid) just before filming. TL transmitted light. Scale bar: 5 μm. **E** *HOF1-3XminiAID* cells were grown in SD medium at 30 °C and treated with 0.1 mM NAA (1-naphtaleneacetic acid) or solvent alone (EtOH, -NAA) for the indicated times. Protein extracts were analysed by western blot with anti-AID antibodies to assess degradation of Hof1-3XminiAID. Pgk1: loading control. The experiment was repeated once more leading to similar results. **F**, **G** Cells with the indicated genotypes and expressing GFP-Cdc12 were grown in SD-raffinose and induced for 90 min with galactose before imaging in SD-raffinose/galactose at 30 °C. Septin splitting and lack of splitting or abnormal splitting events have been quantified in cells undergoing mitotic exit and a new round of budding (**G**). In **C**, **D**, **F** white arrowheads indicate septin splitting; yellow arrowheads indicate a lack of septin splitting (i.e. septin collars that persist after mitotic exit and rebudding). TL transmitted light. Scale bar: 5 μm.

growing in SD medium at 30 °C caused rapid Hof1 depletion (Fig. 1E), suggesting that the system may be suitable for further microscopy-based analyses. It is worth noting, however, that higher doses of NAA are necessary to halt the proliferation of *HOF1-3xmiAID* cells at high temperatures (Fig. S2), suggesting that low levels of Hof1 are sufficient for cell viability.

Given the technical shortcomings that precluded a detailed characterisation of Hof1 contribution to septin ring splitting, we turned to a sensitised experimental setup. Previous work reported that *HOF1* deletion is synthetically lethal with moderate *DMA2* overexpression, to levels that do not otherwise interfere with cell proliferation and cytokinesis[67]. Indeed, while the integration of multiple copies of the *GAL1-DMA2* construct into the genome (hereafter referred to as *GAL1-DMA2m*) prevents septin ring splitting and cytokinesis, integration of one single copy of *GAL1-DMA2* (hereafter referred to as *GAL1-DMA2s*) has very mild effects on cell proliferation[62]. Since *GAL1-DMA2s hof1Δ* cells were shown to delay AMR constriction[67], we asked if this could be the consequence of an underlying defect in septin reorganisation at cytokinesis. To this end, we generated *GAL1-DMA2s HOF1-3xmiAID* cells expressing the fluorescent septin GFP-Cdc12 and analysed them by live cell imaging in the presence of 0.1 mM NAA. While the large majority of *GAL1-DMA2s* or *HOF1-3xmiAID* cells underwent septin splitting/clearance similar to the wild-type control in the presence of galactose, the *GAL1-DMA2s HOF1-3xmiAID* double mutant failed normal septin reorganisation in about 80% of cells, displaying either a septin collar that persisted at the bud neck well after mitotic exit and formation of a new septin ring (35%), or aberrantly rearranged septin collars that lingered at the division site (45%; Fig. 1F, G). Similar data were obtained with *GAL1-DMA2s hof1Δ* mutant cells expressing the fluorescent septin mCherry-Cdc3 (Fig. S3). Therefore, the delay in AMR constriction previously observed in *GAL1-DMA2s hof1Δ* cells[67] may be accounted for by flawed septin ring splitting at cytokinesis.

As a whole, our data indicate that Hof1 contributes to septin ring splitting, although the suboptimal conditions that we had to use for live cell imaging most likely led to an underestimation of the septin splitting defects in Hof1-deleted/depleted cells.

**Mimicking constitutive Hof1 phosphorylation restores septin reorganisation of MEN-defective mutants at cytokinesis**

While previous data suggested that Hof1 stabilises the septin collar[42], the data above indicate that Hof1 may also have a positive role in septin filament disassembly that accompanies septin ring splitting. We hypothesised that the translocation of Hof1 from septins to the AMR, which depends partly on its Dbf2-dependent phosphorylation[37,42], may promote septin reorganisation at cytokinesis. Time-resolved quantitative mass spectrometry[39,68] uncovered several Hof1 phosphorylation sites (S313, S314, S337, T341, T350, S366, S421, S423, S424, S563) arising prior to cytokinesis and being dephosphorylated afterwards (Fig. 2A). Three additional sites (S369, S370 and S478) were gradually dephosphorylated from anaphase to cytokinesis (Fig. 2A). All phosphorylation sites reside in the central intrinsically disordered domain of Hof1 and several, but not all of them, lie in the Septin-Interacting Domain (SID; Fig. 2B). Different mitotic kinases, such as Cdk1, Polo (Cdc5) and Dbf2 are known to phosphorylate these and additional sites[37], while the kinases responsible for phosphorylating S314, S369 and S423 are unknown. Their cell cycle-dependent fluctuations suggest that these phosphorylation/dephosphorylation events may play an important regulatory function. To address this question, we created a series of mutants (*HOF1-P1* to *-P5*) where different sets of residues were mutated into alanine to abolish phosphorylation or to glutamate to mimic constitutive phosphorylation (Fig. 2B). In particular, *HOF1-P1* was mutated in all thirteen mitotically regulated phosphorylation sites that we identified; *HOF1-P2* is mutated in twelve Dbf2-dependent phosphorylation sites and is identical to a previously characterised mutant[37]; *HOF1-P3* is mutated in nineteen known or presumed Cdk1- and Polo-dependent phosphorylation sites; *HOF1-P4* is similar to *HOF1-P1* but is only mutated in the ten sites that are phosphorylated prior to cytokinesis and later dephosphorylated, while *HOF1-P5* is mutated in the remaining three sites that are dephosphorylated during mitotic progression. The slow-migrating isoforms of Hof1-3HA, which are due to Hof1 phosphorylation[37], were only mildly affected in the *HOF1-P1A* and *HOF1-P4A* mutants during the cell cycle (Fig. S4A, B), while they were markedly reduced in *dbf20Δ dbf2-2* mutant cells (Fig. S4C), suggesting that some of the phosphorylations we detected by mass spectrometry may not affect Hof1 electrophoretic behaviour. In contrast, phosphomimetic mutations markedly retarded the electrophoretic mobility of the fast-migrating Hof1 isoforms (Figs. 2C and S4A, B). This mobility shift is unlikely due to new phosphorylations, because it did not collapse after lambda phosphatase treatment (Fig. S5A). Cycloheximide chase assays showed that some mutations (e.g. *HOF1-P4A* and *HOF1-P4E*, Fig. S5B) led to stabilisation of Hof1-3HA, suggesting that they may interfere with Hof1 degradation by the SCF^Grr1 ubiquitin ligase complex[69]. However, this did not appreciably impact the fluctuations of Hof1-3HA levels throughout the cell cycle (Fig. S4A, B).

All *HOF1* mutants proliferated with similar kinetics to the wild-type isogenic strain at all temperatures tested (Fig. S5C). In contrast, only *HOF1-P1E* and *HOF1-P4E* could suppress the lethality of *GAL1-DMA2m* cells on galactose-containing plates (Fig. 2D), suggesting that they might overcome their cytokinetic defects. To address this possibility, we filmed *GAL1-DMA2m* cells carrying the different *HOF1* phosphorylation mutant alleles and expressing mCherry-Cdc3 to follow septin dynamics. While *GAL1-DMA2m* cells failed to undergo septin ring splitting in 70–85% of the cells, the presence of *HOF1-P1E*, *HOF1-P2E* and *HOF1-P4E*, but not their non-phosphorylatable counterparts,

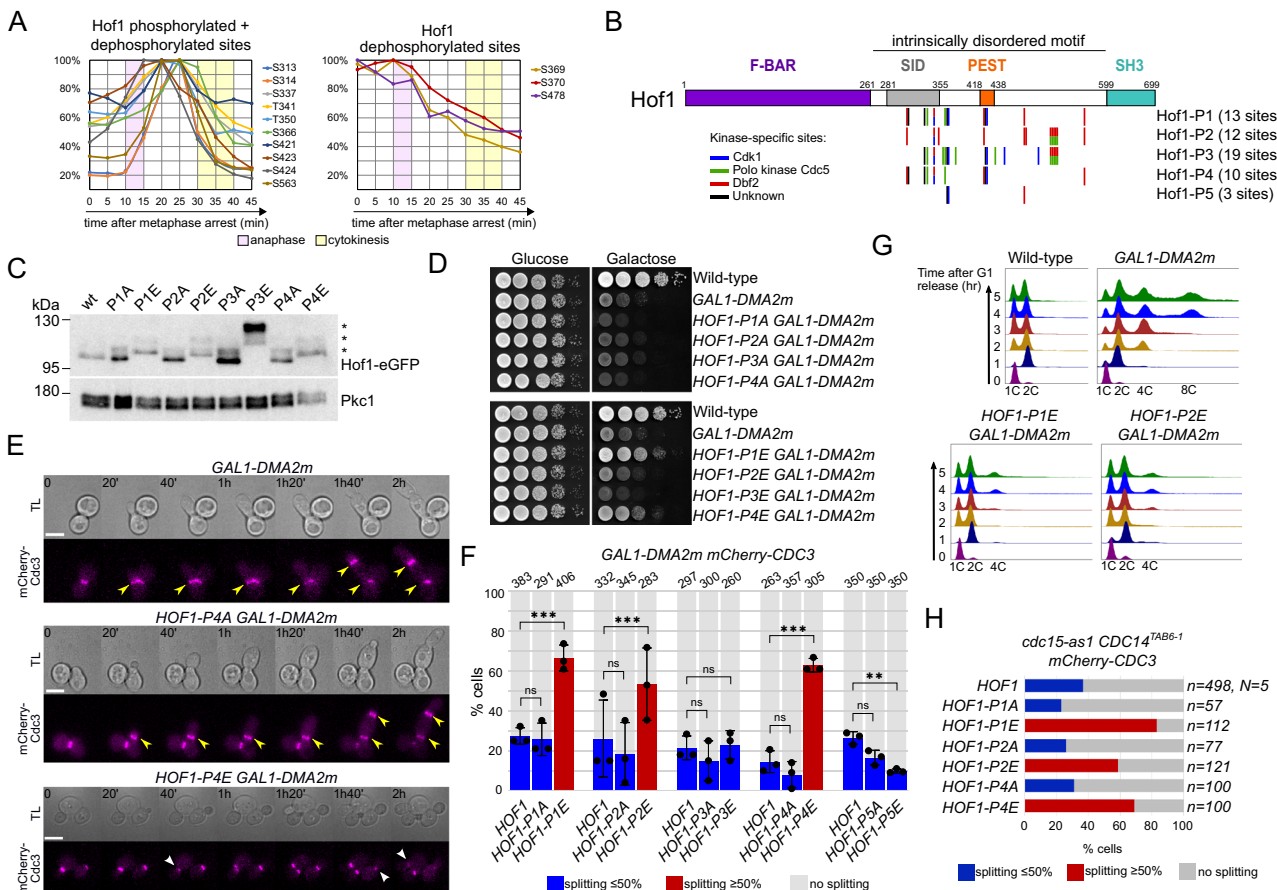

**Fig. 2 | Hof1 phosphorylation promotes septin ring splitting under MEN-inactivating conditions. A** Profiles of Hof1 phosphorylation sites during mitotic progression. For a better visualisation of data, phospho-profiles were plotted relative to the highest value that was considered 100%. The plots represent mean values from independent datasets[39,68]. **B** Schematic representation of Hof1 domains with the position of phosphorylation sites mutated in the different mutant proteins. The colour code indicates the kinases that phosphorylate each site. SID septin-interaction domain. **C** Protein extracts from cells expressing the indicated Hof1 mutant proteins tagged with eGFP were analysed by western blotting with anti-GFP antibodies. Pkc1: loading control. **D** Serial dilutions of cells with the indicated genotypes were spotted on glucose- and galactose-containing YEP plates and incubated at 30 °C. **E, F** *GAL1-DMA2m HOF1* phosphomutants expressing mCherry-Cdc3 were grown in SD-raffinose and induced for 90 min with galactose before imaging in SD-raffinose/galactose at 30 °C. Septin splitting and lack of splitting events have been scored in cells undergoing mitotic exit and a new round of budding during the movie. The mean fraction of cells undergoing septin ring splitting is shown in blue when below 50% and in red when above 50%. Error bars:

SD. The total number of cells analysed per condition is shown at the top of each column (number of biological replicates *N* = 3). *P*-values were calculated with a two-sided $\chi^2$ test. *HOF1* vs *HOF1-P1A*: ns ($p = 0.873$); *HOF1* vs *HOF1-P1E*: *** ($p = 1.45E^{-8}$); *HOF1* vs *HOF1-P2A*: ns ($p = 0.172$); *HOF1* vs *HOF1-P2E*: *** ($p = 9.42E^{-5}$); *HOF1* vs *HOF1-P3A*: ns ($p = 0.269$); *HOF1* vs *HOF1-P3E*: ns ($p = 0.732$); *HOF1* vs *HOF1-P4A*: ns ($p = 0.120$); *HOF1* vs *HOF1-P4E*: *** ($p = 3.44E^{-12}$); *HOF1* vs *HOF1-P5A*: ns ($p = 0.121$); *HOF1* vs *HOF1-P5E*: ** ($p = 0.003$); ns: not significant. Representative *GAL1-DMA2m*, *GAL1-DMA2m HOF1-P4A* and *GAL1-DMA2m HOF1-P4E* cells are shown in (**E**) as an example. Yellow arrowheads indicate septin ring formation in the absence of previous septin ring splitting; white arrowheads indicate normal septin ring splitting/clearance at cytokinesis. TL transmitted light. Scale bar: 5 μm. **G** Cells with the indicated genotypes were grown in YEPR, arrested in G1 by α-factor (*t* = 0) and induced with 1% galactose 30 min before release in YEPRG. At the indicated times cells were withdrawn for flow-cytometric analysis of DNA contents. **H** *cdc15-as1 CDC14^{TAB6-1}* cells carrying the indicated *HOF1* phospho-mutant alleles and expressing mCherry-Cdc3 were filmed in SD medium at 30 °C in the presence of 5 μM 1NM-PP1. Septin splitting and lack of splitting events have been quantified as in (**F**).

could restore septin splitting/clearance in the majority of cells (Fig. 2E, F). *HOF1-P3* and *HOF1-P5*, either in the A or E version, had no ameliorating effect in this context and were not analysed further. Flow-cytometric analysis of DNA contents of cells synchronised in G1 and released in the presence of galactose confirmed that while *GAL1-DMA2m* cells experienced severe cytokinetic defects and accumulated 4C–8C DNA contents at 4–5 h after release, the presence of *HOF1-P1E* and *HOF1-P2E*, but not their non-phosphorylatable counterparts, could mostly rescue these defects and re-establish a bimodal 1C–2C distribution of DNA contents similar to wild type (Figs. 2G and S5D). The reason why *HOF1-P4E* did not seem to restore normal DNA profiles under these conditions is unclear (Fig. S5D), but it could be due to problems in primary septum formation that is known to be controlled by Hof1[43,56].

To exclude that the effects of the *HOF1* phosphomimicking mutants were specific to *GAL1-DMA2* cells, we tested the effects of the *HOF1* mutant alleles in another yeast mutant that fails septin ring splitting because of impaired MEN activity. To this end, we used the analogue-sensitive *cdc15-as1 CDC14^{TAB6-1}* mutant, which in the presence of the 1NM-PP1 ATP analogue is defective in septin ring splitting despite being able to exit mitosis due to MEN-independent Cdc14 activation[28]. The presence of the *HOF1-P1E*, *-P2E* and *-P4E* alleles, but not their non-phosphorylatable counterparts, was sufficient to restore septin splitting/clearance at the bud neck in the majority of cells (Fig. 2H), confirming that mimicking specific Hof1 phosphorylations by Dbf2 and other unidentified kinases can bring about septin reorganisation independently of MEN. In contrast, phosphorylations by Polo kinase and Cdk1 do not seem to play any role in this process.

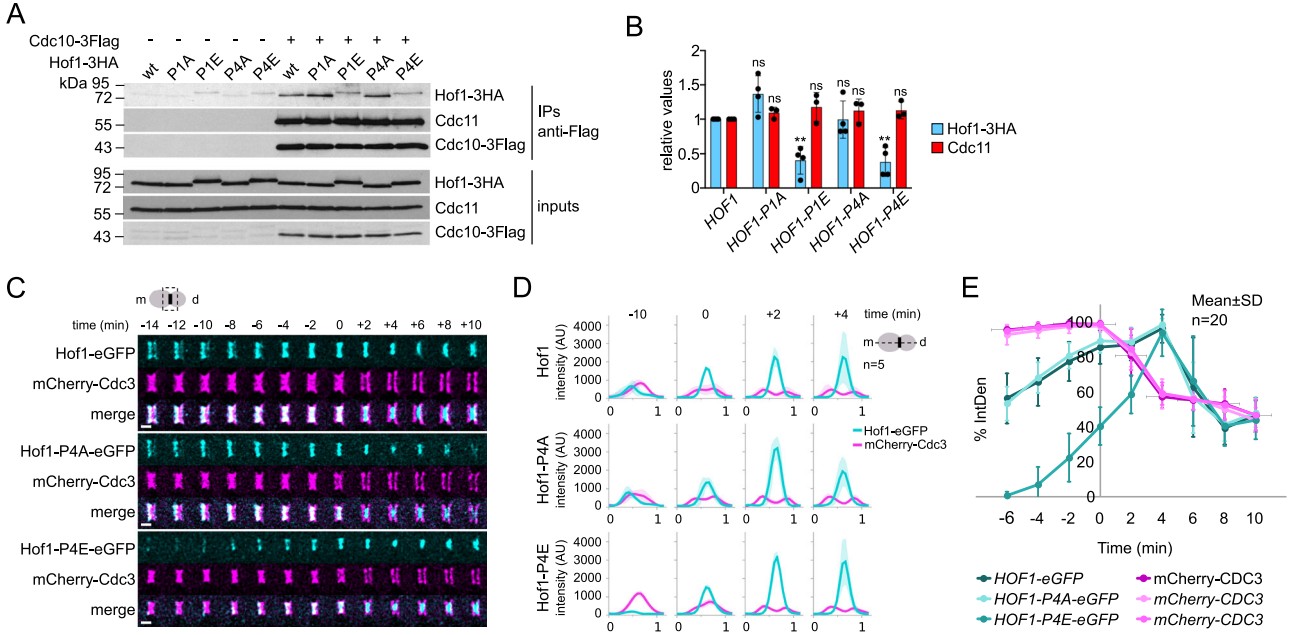

**Fig. 3 | *HOF1* phosphomimicking mutations affect Hof1 binding to septins.**
**A, B** Cells expressing either untagged or Flag-tagged *CDC10* (Cdc10-3Flag) were arrested in mitosis by nocodazole. Protein extracts were subjected to immuno-precipitations with an anti-Flag antibody; immunoprecipitates and inputs were probed by western blot with anti-HA, anti-Flag and anti-Cdc11 antibodies. Inputs represent 1/60th of the extracts used for IPs. Amounts of Hof1-3HA and Cdc11 co-immunoprecipitated with Cdc10-3Flag were quantified with ImageJ in three (Cdc11) or four (Hof1-3HA) independent experiments; mean values relative to the wild-type control, which was arbitrarily set to 1, were plotted in (**B**). Error bars: SD. *P*-values were calculated with an unpaired two-tailed *t*-test. *HOF1* vs *HOF1-P1A*: Hof1-3HA ns (*p* = 0.07), Cdc11 ns (*p* = 0.184); *HOF1* vs *HOF1-P1E*: Hof1-3HA ** (*p* = 0.0096), Cdc11 ns (*p* = 0.28); *HOF1* vs *HOF1-P4A*: Hof1-3HA ns (*p* = 0.977), Cdc11 ns (*p* = 0.334); *HOF1*

vs *HOF1-P4E*: Hof1-3HA ** (*p* = 0.0099), Cdc11 ns (*p* = 0.214); ns: not significant. **C, D** Cells expressing mCherry-Cdc3 and the indicated Hof1-eGFP proteins were filmed in SD medium at 30 °C. Images were deconvolved with the Andor Fusion software to resolve single and double Hof1 rings. Time 0 is the time of septin ring splitting. TL transmitted light. Scale bar: 1 μm. Graphs in (**D**) represent Hof1-eGFP and mCherry-Cdc3 average fluorescence intensities (*n* = 5) along a line spanning the bud neck. Shaded curves: SD. **E** Cells expressing mCherry-Cdc3 and different Hof1-eGFP protein variants were imaged in SD medium at 30 °C. Fluorescence intensities at the bud neck were normalised to the highest value, which was arbitrarily set at 100%, averaged (*n* = 20) and plotted. Time 0 corresponds to the frame immediately before septin ring splitting. Errors bars: SD.

## Hof1 phosphorylation weakens Hof1 interaction with septins and promotes its translocation to the AMR

The Hof1-P2E mutant protein was previously shown to bind less efficiently to septins than wild-type Hof1 and to relocate from the septin collar to the medial AMR in telophase partly independently of MEN[37]. This prompted us to test the affinity of the Hof1-P1E and -P4E proteins for septins and their localisation at the bud neck. Immunoprecipitation of the Flag-tagged septin Cdc10 from mitotically arrested cells could co-precipitate wild-type Hof1 as well as the non-phosphorylatable Hof1-P1A and -P4A, but significantly lower levels of the phosphomimetic Hof1-P1E and -P4E (Fig. 3A, B), suggesting that Hof1 phosphorylation by MEN reduces its affinity for septins. Consistent with a role of Hof1 in septin collar stability[42], *HOF1-P1E* and *-P2E* displayed severe growth defects in combination with deletion of the only non-essential septin Shs1, while the *-P1A* and *-P2A* counterparts had no notable effect (Fig. S6A).

Hof1 is recruited asymmetrically to the edges of the septin collar in mitosis, appearing first at its mother side and afterwards transiently at its bud side, shortly before translocating to the AMR at the time of septin ring splitting as a single ring capable of partial constriction[37,41,43,58,59] (Figs. 3C, D and 1C). In line with the conclusion that the *HOF1-P1E* and *-P4E* phosphomimetic mutants hamper Hof1 interaction with septins, recruitment of eGFP-tagged Hof1-P4E (Fig. 3C–E) and -P1E (Fig. S6B, C) to the septin collar was severely impaired during mitosis (i.e. before septin ring splitting), while the same proteins could localise as efficiently as wild-type Hof1 at the AMR at cytokinesis. In stark contrast, the non-phosphorylatable Hof1-P4A and -P1A mutant protein behaved indistinguishably from their wild-

type counterpart (Figs. 3C–E and S6B, C). Thus, Hof1 phosphorylation at mitotic exit unleashes Hof1 from septins, thereby allowing its translocation to the contractile AMR, as previously proposed[37]. It should be noticed, however, that phosphomimetic Hof1 mutant proteins do not prematurely localise to the AMR, suggesting that their so far unidentified interactor(s) at the AMR might be absent from the bud neck before cytokinesis. Experiments designed to evaluate if and how *DMA2* overexpression affects Hof1 localisation at the bud neck were not conclusive. Indeed, Hof1 was unexpectedly expressed at higher levels in *GAL1-DMA2* than in wild-type cells during mitosis (Fig. S7A) and abnormally localised on the whole septin collar during mitosis (Fig. S7B), which precluded the assessment of its possible relocalisation to the AMR at mitotic exit. This behaviour was reminiscent of a stabilised Hof1 protein lacking its PEST domain[69]. However, the reason why Hof1 may be stabilised upon *DMA2* overexpression is unclear at the moment and will be investigated in the future.

## The F-BAR domain of Hof1 is involved in its function in septin ring splitting

Finding that Hof1 promotes septin ring splitting raised the possibility that Hof1 translocation to the AMR may cause a modification of the plasma membrane at the division site incompatible with septin assemblies. The F-BAR domain of Hof1 would be perfectly suited for this role, as F-BAR domains bind to phospholipids and are able to sense and/or induce membrane curvature[70,71]. We therefore deleted the F-BAR domain (aa 2–279) from *HOF1*, either wild-type or phosphomimetic variants, to assess its contribution to septin ring splitting. Yeast mutants lacking the Hof1 F-BAR domain could proliferate like the

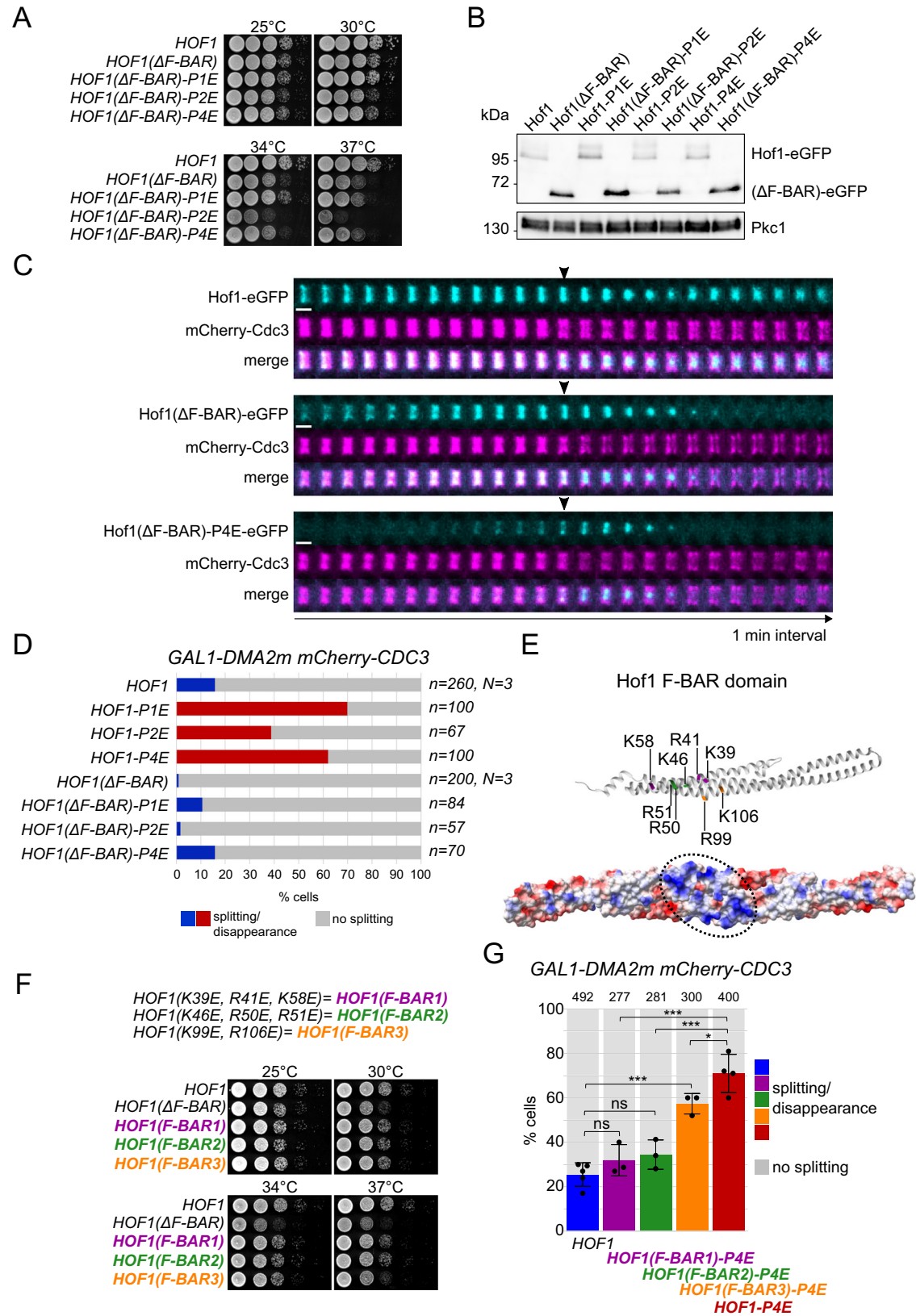

wild-type control at 25–30 °C, while they were slightly (*HOF1, HOF1-P1E* and *HOF1-P4E*) or severely (*HOF1-P2E*) temperature-sensitive at 34–37 °C (Fig. 4A). Deletion of the F-BAR domain did not reduce Hof1 protein levels (Fig. 4B). Furthermore, time-lapse imaging showed that the Hof1(ΔF-BAR)-eGFP protein was recruited to septins in mitosis and shifted to the AMR with similar kinetics as Hof1-eGFP (Fig. 4C). One

notable difference, however, was that Hof1(ΔF-BAR)-eGFP seemed to constrict to a higher extent than wild-type Hof1-eGFP and to leave the division site afterwards, while Hof1-eGFP after partial constriction redistributed to the two sides of the division site, as previously shown[41,43]. Very low levels of the phosphomimetic Hof1(ΔF-BAR)-P4E-eGFP were found at the septin collar, similar to what we observed for

**Fig. 4 | The F-BAR domain is required for the septin splitting function of Hof1.**
**A** Serial dilutions of cells with the indicated genotypes were spotted on YEPD and incubated at the indicated temperatures. **B** Protein levels of the indicated Hof1 variants tagged with eGFP were analysed by western blot with anti-GFP antibodies. Pkc1: loading control. **C** Cells expressing mCherry-Cdc3 and the indicated Hof1-eGFP proteins were filmed in SD medium at 30 °C. Arrowheads indicate the time of septin ring splitting. Scale bar: 1 μm. **D** *GAL1-DMA2m mCherry-CDC3* cells carrying the indicated *HOF1* alleles were grown in SD-raffinose and induced for 90 min with galactose before imaging in SD-raffinose/galactose at 30 °C. Septin splitting and lack of splitting events have been quantified in cells undergoing mitotic exit and a new round of budding. The fraction of cells undergoing septin ring splitting is shown in red when at least two-fold higher than in the control *GAL1-DMA2m HOF1* strain. **E** Top: Structure of the monomeric Hof1 F-BAR domain[73]. Basic residues that have been mutated are highlighted (top). The colour code corresponds to the mutants in (**F, G**). Bottom: distribution of the electrostatic potential on the concave surface of dimeric Hof1 F-BAR domain. Blue: positive potential; red: negative potential; white: near neutral. The major basic patch bearing the mutated residues is encircled. **F** Serial dilutions of cells with the indicated genotypes were spotted on YEPD and incubated at the indicated temperatures. **G** *GAL1-DMA2m* cells bearing the indicated *HOF1* alleles and expressing mCherry-Cdc3 were grown in SD-raffinose and induced for 90 min with galactose before imaging in SD-raffinose/galactose at 30 °C. Septin splitting and lack of splitting events have been scored in cells undergoing mitotic exit and a new round of budding during the movie. The mean fraction of cells undergoing septin ring splitting is shown. Error bars: SD. The total number of cells analysed per condition is shown at the top of each column (number of biological replicates N = 3). *P*-values were calculated with a two-sided $\chi^2$ test. *HOF1* vs *HOF1(F-BAR1)-P4E*: ns (*p* = 0.273); *HOF1* vs *HOF1(F-BAR2)-P4E*: ns (*p* = 0.163); *HOF1* vs *HOF1(F-BAR3)-P4E*: *** (*p* = 4.2E⁻⁶); *HOF1(F-BAR1)-P4E* vs *HOF1-P4E*: *** (*p* = 3.43E⁻⁸); *HOF1(F-BAR2)-P4E* vs *HOF1-P4E*: *** (*p* = 1.61E⁻⁷); *HOF1(F-BAR3)-P4E* vs *HOF1-P4E*: * (*p* = 0.039); ns: not significant.

Hof1-P4E-eGFP, while the protein peaked at the bud neck around the time of septin ring splitting and constricted similar to Hof1(ΔF-BAR)-eGFP (Fig. 4C). Thus, deletion of the F-BAR domain does not substantially affect the kinetics of Hof1 interaction with the septin collar or the AMR in otherwise wild-type cells.

To assess the possible involvement of the Hof1 F-BAR domain in septin ring splitting, we introduced the *ΔF-BAR* mutant alleles into *GAL1-DMA2m* cells expressing mCherry-Cdc3. Strikingly, the lack of the F-BAR domain abolished the ability of *HOF1-P1E*, *-P2E* and *-P4E* to rescue the septin splitting defects due to *DMA2* overexpression (Fig. 4D). Furthermore, eliminating the F-BAR domain in wild-type Hof1 caused additive septin splitting defects with *DMA2* overexpression (Fig. 4D).

F-BAR domains are thought to bind and bend membranes via their concave surface, which bears clusters of positively charged amino acids that engage electrostatic interactions with membrane phosphoinositides[72]. To confirm the involvement of Hof1 F-BAR domain in promoting septin ring splitting, we generated three *HOF1* mutant alleles (*HOF1(F-BAR1)*, *HOF1(F-BAR2)* and *HOF1(F-BAR3)*) in which positively charged residues that are exposed on the concave surface of Hof1 F-BAR domain (K39/R41/K58; K46/R50/R51; R99/K106, respectively[73]; Fig. 4E) were mutated into opposite charged glutamate. Basic residues corresponding to K39 and R41 in the human F-BAR protein FBP17 (K33 and R35) were previously implicated in membrane binding and tubulation[74,75]. *HOF1* F-BAR mutations did not impact cell viability at 25 °C, while they caused a slight temperature-sensitive growth phenotype that was less severe than that elicited by deletion of the entire F-BAR domain (Fig. 4F). Importantly, when these mutations were introduced into the *HOF1-4E* allele, they significantly hampered the ability of Hof1-4E to suppress the septin splitting defects of *GAL1-DMA2m* cells, with *HOF1(F-BAR1)* and *HOF1(F-BAR2)* having the strongest effect (Fig. 4G).

Thus, the Hof1 F-BAR domain is an important determinant for septin ring splitting, suggesting that it might induce a local rearrangement of the plasma membrane at the division site that contributes to septin reorganisation at cytokinesis.

## Hof1 bundles septin filaments into large networks in vitro

In order to investigate if and how Hof1 modulates septin assembly, we used in vitro reconstitution assays[76] using recombinant Cdc11-Cdc12-Cdc3-Cdc10-Cdc10-Cdc3-Cdc12-Cdc11 septin octamers[77] and MBP-tagged Hof1[53], either wild-type or phosphomimetic -P4E. Dynamic light scattering (DLS) indicated that septins polymerising in low salt buffer (30 mM NaCl) were able to rapidly form big assemblies in the presence of equimolar amounts of Hof1 or, to a lower extent, Hof1-P4E (Fig. 5A). Consistently, upon lowering the salt concentration, eGFP-labelled septin octamers were able to polymerise into long filaments (around 1–2 μm), in agreement with previous data[78], while they organised into large networks in the presence of equimolar amounts of Hof1 (Fig. 5B). This is in agreement with the recently reported septin-

bundling activity of Hof1[53]. Importantly, Hof1 tagged with blue fluorescent protein (BFP) colocalised with septin bundles and networks (Fig. S8A), indicating that it is part of septin assemblies.

The addition of Hof1-P4E had a less prominent effect than Hof1, with smaller structures forming on a layer of septin filaments (Fig. 5B), while Hof1-4A could organise septins into supermolecular structures as efficiently as wild-type Hof1 (Fig. S8B). Deletion of either the septin-interacting domain (SID[42,79]) or the F-BAR domain that also participates to septin interaction[37] significantly hampered the ability of Hof1 to bundle and organise septins (Fig. S8B).

To visualise septin assemblies at high resolution, we used transmission electron microscopy (TEM) after negative staining. As expected, septins alone formed long paired filaments, while Hof1 alone, either wild type or -P4E, did not polymerise. Interestingly, Hof1 particles appeared more globular and compact than Hof1-P4E (Fig. 5C), raising the possibility that mimicking Hof1 phosphorylation might induce a conformational change from a "closed" to more "open" configuration, similar to what has been shown for its Cdc15 ortholog in fission yeast[80,81]. The addition of scaling amounts of Hof1 to septins during polymerisation caused the formation of big, interlaced meshes of septin filaments and bundles (Fig. 5D–F) that often contained thick nodes of heavily stained dense material (blue arrows). At increasing magnifications, septin bundles appeared to be made by several septin filaments aligned side-by-side with lateral crosslinks (pink arrowheads) that could represent Hof1 itself.

Hof1-P4E induced the formation of similar septin networks, although much less efficiently than wild-type Hof1, since higher protein ratios were necessary to induce higher-order septin organisation (Fig. 5D–F). This is consistent with our DLS data and with the reduced binding of Hof1-P4E to septins in vivo. Septin filaments polymerised in the presence of Hof1-P4E also appeared more curved and twisted relative to those assembled in the presence of Hof1.

To investigate how Hof1 impacts septin organisation on membranes, we studied septin assembly on flat-supported lipid bilayers (SLBs) using fluorescent eGFP-tagged yeast septin octamers. Since yeast septins bind preferentially phosphatidylinositol(4,5)-bisphosphate (hereafter referred to as PI(4,5)P₂)[14,17], we doped our biomimetic membranes with 3% PI(4,5)P₂ (including 0.2% mol of TopFluor-TMR-PI(4,5)P₂ to visualise the membrane plane). Septins could form small assemblies on SLBs that we interpret as short filaments, although the resolution was insufficient to clearly visualise them (Fig. 6A). Addition of different molar amounts of Hof1 drastically reshaped septin assemblies, inducing the formation of huge bundles and meshes partly adherent to the membrane and partly in the solution above the bilayer, in agreement with the ability of Hof1 to bundle septin filaments also in solution (Figs. 6B and S9A, B). Compact nodes reminiscent of the dense structures we observed by TEM were also apparent (arrows). Septin bundling by Hof1 was fast: immediately after Hof1 addition to septins we could observe patches of septins coalescing and

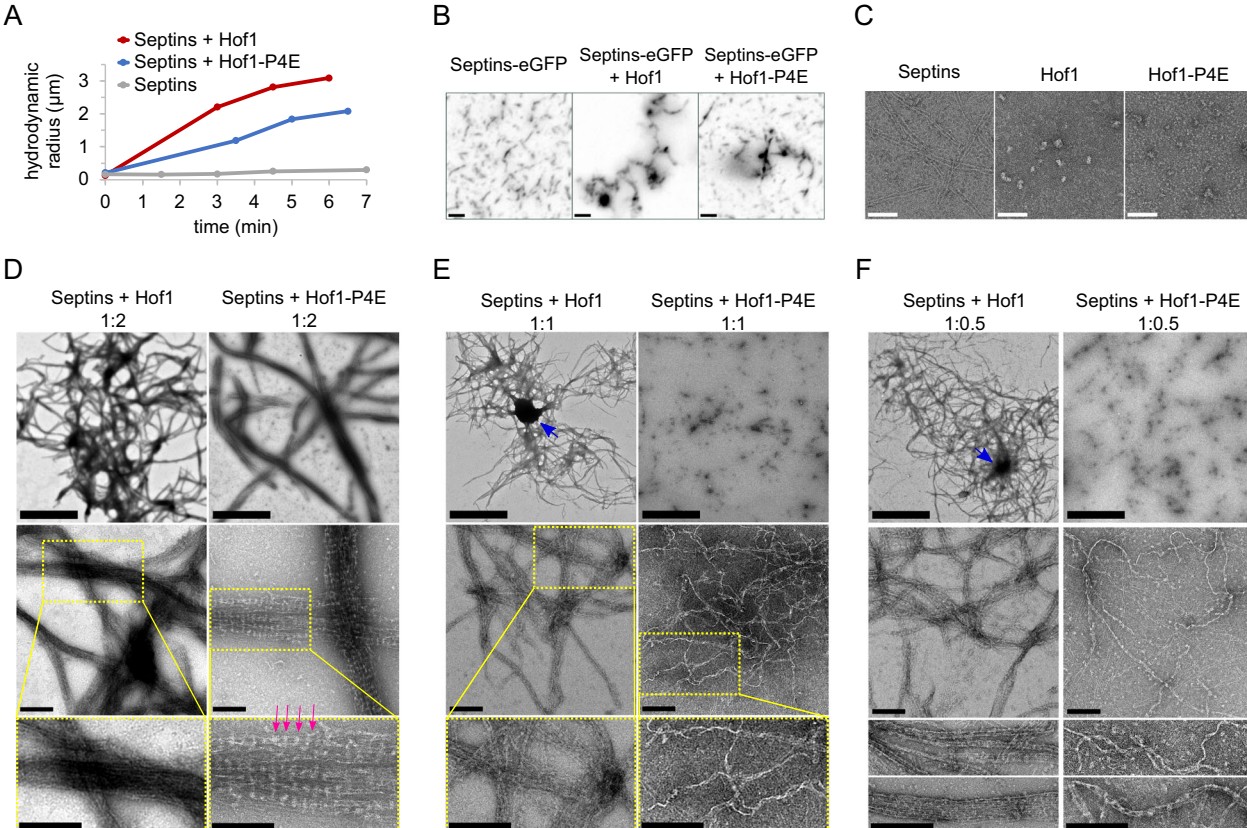

**Fig. 5 | Hof1 induces septin higher-order structures in solution. A** 180 nM of recombinant Cdc11-capped septin octamers were polymerised in solution by lowering the salt concentration and analysed by conventional DLS at a scattering angle of 90° in the absence of Hof1 or after the addition of equimolar amounts of wild-type Hof1 or the phosphomimicking Hof1-P4E protein ($t = 0$). **B** A 1:1 mix of untagged and eGFP-tagged Cdc11-capped septin octamers (50 nM final concentration) were polymerised in solution by lowering the salt concentration in the absence or in the presence of equimolar amounts of Hof1 or Hof1-P4E and then imaged by fluorescence microscopy. Scale bar: 2 µm. The experiments were repeated once leading to similar results. **C** Purified Cdc11-capped septin octamers (180 nM) and Hof1 proteins (360 nM) were diluted with low salt buffer and imaged by TEM. Scale bar: 100 nm. The experiments were repeated twice leading to similar results. **D–F** Recombinant Cdc11-capped septin octamers (180 nM) were polymerised in solution by lowering the salt concentration in the presence of the indicated molar ratios of Hof1 or Hof1-P4E. Images show septin assemblies at different magnifications (scale bar of top images 2 µm; scale bar of middle and bottom images: 200 nm). Insets highlighted by yellow dashed rectangles are magnified in the bottom images. Magenta arrowheads indicate periodic banding of septin bundles. Blue arrows indicate thick nodes of compact electron-dense material that could be due to tight septin packing. The experiments were repeated once (**D**) or twice (**E, F**) leading to similar results.

transforming into thick bundles with time (Fig. S8A). 3D-reconstruction revealed that the Hof1-induced septin meshwork could span a thickness of more than 29 µm above the SLB (Fig. 6B). Notably, the presence of PI(4,5)P$_2$ was not necessary for Hof1 to promote septin bundling and intertwining on membranes. Indeed, we could get similar results with SLBs made of net-neutral DOPC and containing 4% DGS-NTA(Ni) lipids to recruit septin complexes (that were 6His-tagged for purification; Fig. S9D), suggesting that the ability of Hof1 to bundle septins is independent of phospholipids. In stark contrast to wild-type Hof1, the phosphomimetic Hof1-P4E, even at high molar ratios relative to septins, could only promote the assembly of sparse and relatively small septin assemblies, again confirming that it seriously impairs the septin-organising activity of Hof1 (Figs. 6A–C; S9C, D).

Altogether, these data indicate that Hof1 prompts the formation of septin higher-order structures. Hof1 phosphorylation interferes with this function likely by decreasing Hof1 affinity for septins.

## Discussion

Reorganisation of the septin scaffold at the mitotic exit, through septin ring splitting or disassembly, is a compulsory step for budding yeast cytokinesis. Indeed, septins must clear the division site to allow AMR constriction, presumably because septins may locally rigidify the membrane or obstruct productive membrane-AMR interactions[82].

In this paper, we uncover a novel function for the cytokinetic protein Hof1 in septin ring splitting. Hof1 binds the septin collar in mitosis and here we show that it is able to bundle septin filaments, arranging them into large networks. The septin-bundling activity of Hof1 may contribute to the rigidity of the septin collar and septin stability during mitosis, as previously proposed[42]. In this framework, MEN-driven relocalisation of Hof1 from septins to the AMR could on its own destabilise the septin collar and contribute to septin disassembly at the double ring transition. However, our data based on *HOF1* deletion/depletion are inconsistent with this simple hypothesis and indicate that Hof1 actively promotes septin ring splitting during its translocation to the AMR. Additionally, we show that the F-BAR domain of Hof1 is critically involved in this process, suggesting that Hof1 interaction with the membrane may be required for Hof1-mediated septin remodelling. Thus, we envision that Hof1 leaves the septin collar to allow septin destabilisation at cytokinesis, but must concomitantly join the medial membrane and the AMR to promote septin rearrangement. Since BAR domains can induce membrane curvature[44,45], one possibility is that Hof1 may curve the membrane at the division site to a radius that is incompatible with septin assemblies. BAR domains have also been shown to cluster phosphoinositides[83,84].

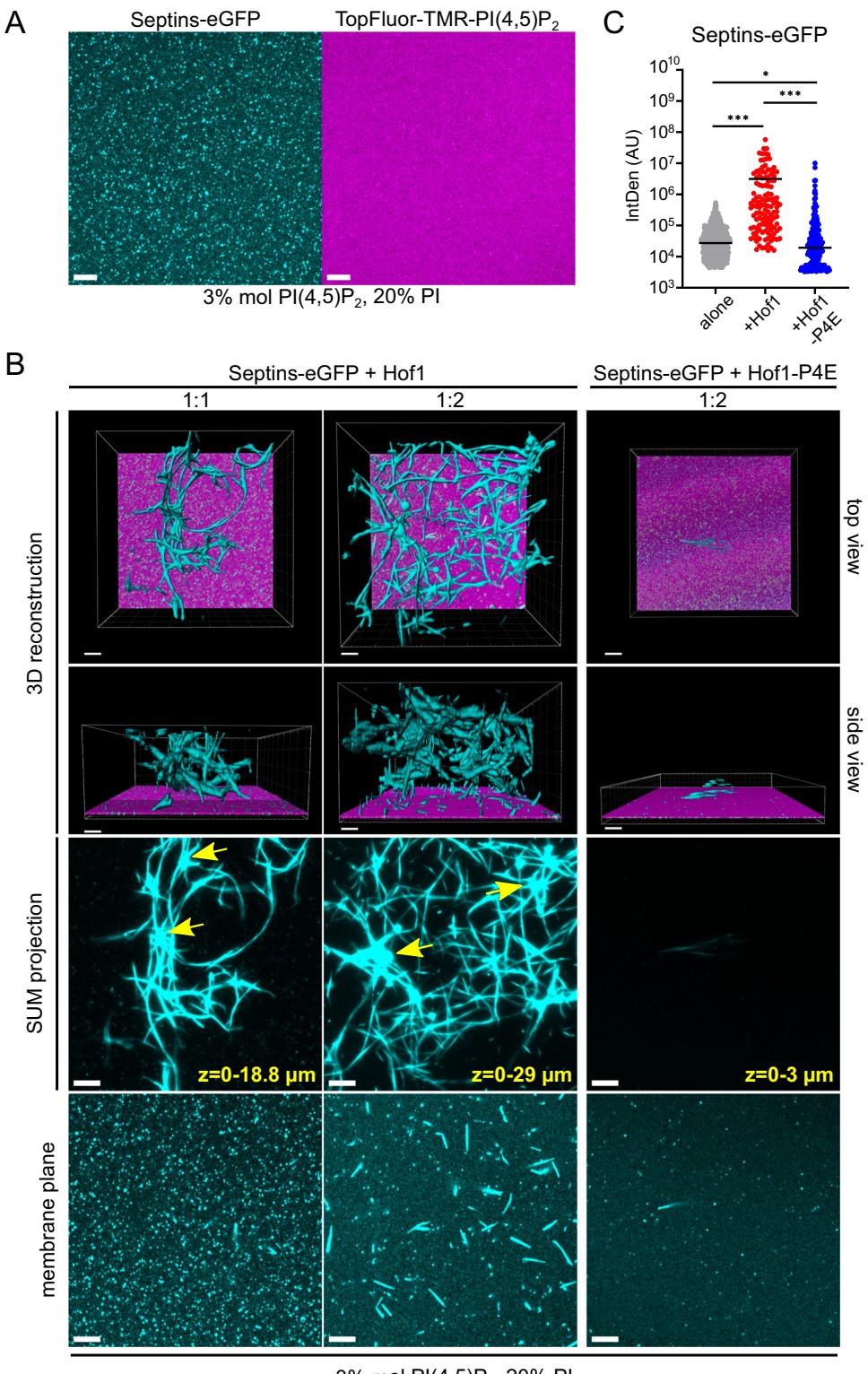

**Fig. 6 | Septins form micrometric networks on membranes in the presence of Hof1. A, B** A 1:1 mix of untagged and GFP-tagged septin octamers (50 nM final concentration) was polymerised in solution before being injected onto SLBs doped with PI(4,5)P₂ and imaged by sub-diffraction Airyscan microscopy. The indicated molar ratios of Hof1 (wild-type or phosphomimicking Hof1-P4E) were added shortly afterwards onto the SLBs. Traces of a fluorescent lipid (TopFluor-TMR-PI(4,5)P2) were used to visualise the plane of the membrane (shown in **B** in magenta). One plane at the membrane level shows septins laid on it, while several planes on the top of the membrane were sum-projected to show septin structures in solution. 3D-image rendering in (**B**) was obtained with the Imaris software. Scale bar: 5 μm. **C** Quantification of the fluorescence associated with septin structures measured with the 3D object counter plugin of ImageJ on $n = 4$ volumes of $20 \times 20 \times 6.5$ μm. *P*-values were calculated with an unpaired two-tailed *t*-test. ****p = 7.08E^{−6}, *p = 2.64E^{−2}, ***p = 2.13E^{−5}, respectively; $n = 596, 126$ and $181$, respectively.

Thus, Hof1 may promote septin ring splitting by changing the local composition of the plasma membrane, thereby decreasing septin affinity for the cleavage furrow. Finally, BAR domains mediate protein dimerisation/oligomerization[85], suggesting that the structural properties of Hof1 may underlie its function in septin ring splitting. Interestingly, *S.p.* Cdc15 is unable to curve membranes, and its oligomerization properties mediated by its F-BAR domain have been involved in its cytokinetic function(s)[86]. Similarly, the F-BAR domain of *S.c.* Hof1 did not induce membrane tabulation when expressed in cultured cells[87]. Whether oligomerisation or other properties of the Hof1 F-BAR domain are involved in septin ring splitting remains to be investigated.

We have identified several phosphorylation sites that modify Hof1 specifically at the time of cytokinesis and facilitate its translocation from septins to the AMR by loosening its association with the septin collar. Some of these sites were previously shown to be phosphorylated by the MEN kinase Mob1-Dbf2[37], while others are phosphorylated by other kinases, whose identity will be explored in the future. A few phosphorylation sites (S313, T350, S421, S563, all expected phosphorylation targets of Mob1-Dbf2[37]) are mutated in all *HOF1* phosphomimetic mutants that positively impact septin ring splitting and might therefore have a prominent role in Hof1 relocalisation. Indeed, it is unlikely that Hof1 translocation to the AMR is simply driven by negative charges in the disordered domain, because phosphomimetic mutations in the CDK and Polo phosphorylation sites cause a marked electrophoretic shift of the protein without affecting its ability to induce septin remodelling. Thus, we favour the idea that phosphorylation of specific residues is key to this process.

We additionally show that *HOF1* phosphomimetic mutants that facilitate septin ring splitting also impair Hof1 interaction with septins in vivo and in vitro, in agreement with previous data[37]. We hypothesise that the same mutations may concomitantly increase Hof1 affinity for the AMR. How exactly Hof1 binds the AMR is not known, but its interaction with IQGAP (Iqg1)[88] and myosin II (Myo1)[58] may be key for this purpose. Interestingly, our *HOF1-P1, -P2* and *-P4* phosphomimetic mutants do not cause premature Hof1 relocalisation to the AMR in unperturbed conditions, suggesting that Hof1 phosphorylation is necessary but not sufficient for this process or that the glutamate mutations do not fully mimic the effects of phosphorylation. Consistently, these mutants are perfectly viable and fit, in contrast to what one would expect in case of unscheduled septin disassembly. Similarly, non-phosphorylatable mutants (*HOF1-P1A, -P2A* and *-P4A*) are also fit and do not affect cytokinesis even under sensitised conditions (i.e. *GAL1-DMA2s*[62]), suggesting that alternative mechanisms must operate to ensure that Hof1 joins the AMR at the right time. Additionally, it should be noticed that none of our *HOF1* alanine mutants completely abolishes Hof1 phosphorylation, thus providing a simple explanation for their lack of phenotypes. Finally, it is also possible that the mutated Hof1-P1A and Hof1-P4A proteins are phosphorylated on secondary sites that are only poorly phosphorylated under physiological conditions.

Our TEM data suggest that mimicking Hof1 phosphorylation may cause a conformational shift from a compact, closed conformation to a more open configuration that could expose buried binding sites. Notably, *S.p.* Cdc15 has been shown to become competent to join the AMR upon a closed-to-open structural rearrangement prompted by protein dephosphorylation, rather than phosphorylation[80]. Thus, although a conformational shift may be required for both Hof1 and Cdc15 to incorporate into the contractile ring, the molecular mechanism to reach this goal seems to diametrically differ in the two yeasts.

Although our data clearly show that *HOF1* deletion/depletion leads to septin splitting defects, the non-essentiality of *HOF1*, especially in some strain background, suggests that other proteins participate to septin ring splitting. Several proteins land at the division site approximately at the time of septin ring splitting and might therefore be instrumental to this process. Proteins whose inactivation is synthetically lethal with *hof1* deletion (i.e. Cyk3, Chs2, Rvs167, etc.[64,89,90]) are promising candidates.

Similar to budding yeast, a septin double ring is formed during cytokinesis in mammalian cells, where it recruits proteins of the ESCRTIII abscission machinery to the division site[91]. Remarkably, septin clearance from the cleavage furrow seems to accompany the abscission process[91], suggesting that septins may have both a positive and a negative impact on cell division in different organisms. In contrast, in fission yeast septins are not required for cytokinesis, but rather for cell separation[92], the process that occurs after cytokinesis and that involves the enzymatic digestion of the primary septum to allow the dividing cells to break apart. In this organism, septins form a ring at the division site only after AMR assembly[93,94]. However, the septin collar is converted into a double ring also in fission yeast cells[93–95], although the functional relevance of this reorganisation is not understood. Whether the formation of the septin double ring depends on the MEN-like Septation Initiation Network or on Cdc15 and whether this septin rearrangement is also necessary for cell separation in fission yeast are intriguing questions for future investigation.

## Methods

### Strains, growth conditions, primers and plasmids

All yeast strains (Table S1) are congenic to W303 (*ade2-1, trp1-1, leu2-3,112, his3-11, 15 ura3*). W303 bears a single nucleotide deletion in the *BUD4* gene (*bud4-G2459fs*) that results in a premature stop codon. The *bud4-G2459fs* gene produces a truncated protein of 838 amino acids that lacks 609 amino acids and carries 18 non-natural amino acids at C-terminus (https://www.yeastgenome.org). This mutation leads to septin ring disassembly, instead of splitting, at mitotic exit[96]. In order to properly visualise septin ring splitting, the genotype of most strains has been corrected to carry full-length *BUD4*[28,97]. Almost all experiments have been performed with the corrected *BUD4+* strains, except those in Figs. S1A, C, S5C.

Yeast cultures were grown at 30 °C in either synthetic medium (SD) supplemented with the appropriate nutrients or YEP (1% yeast extract, 2% bactopeptone, 50 mg/l adenine) medium. Raffinose was supplemented to 2%, glucose to 2% and galactose to 1%. NAA (1-naphthaleneacetic acid) and IAA (3-indoleacetic acid) were dissolved in ethanol as 1000× and 500× stock solutions, respectively, and used at a final concentration 0.1 mM and 1 mM, respectively, unless otherwise indicated. Cells were synchronised in G1 by alpha factor (4 μg/ml) in YEP medium containing the appropriate sugar at 25 °C. G1 arrest was monitored under a transmitted light microscope and cells were released in fresh medium at 30 °C (typically after 120–135 min of alpha factor treatment) after being collected by centrifugation at 2000 × *g* and washed with YEP containing the appropriate sugar. Nocodazole was used at 15 μg/ml. Cycloheximide was used at 1 mg/ml.

One-step tagging techniques were used to generate strains bearing *HOF1* deletion, *HOF1-3XminiAID*, eGFP- and 3HA-tagged *HOF1*, 6Gly-3Flag-tagged *CDC10*, *K.l.TRP1*-marked *RCH1*, as well as to introduce the *kanMX* gene marker upstream or downstream of *HOF1* for CRISPR/Cas9 gene editing. All *HOF1-3XminiAID* strains also express the *O.s.TIR1* gene from the *ADH1* promoter to induce protein degradation[98].

To generate the *HOF1* phosphomutants, as well as *HOF1-(ΔF-BAR)* (Δ2–279) and *HOF1(F-BAR1-3)* mutants by CRISPR/Cas9 gene editing, pSP1712 (Cas9 and gRNA against *kanMX* co-expression vector) and a double-stranded DNA repair template were co-transformed into yeast cells bearing the *kanMX* marker upstream or downstream of *HOF1*. DNA repair templates were amplified from synthetic DNAs cloned in pTwist plasmids (Twist Bioscience). After transformation, edited cells were cured from the co-expression vector and gene editing was confirmed by PCR followed by restriction enzyme digestion and sequencing.

The plasmids used for septin purification were described before[77]. The plasmid used for the expression of MBP-Hof1-6His in *E. coli* (1–669; pBG2163) was a gift from B. Goode (Brandeis University, Waltham, MA) and has been previously described[54]. The plasmids for expression of MBP-Hof1(ΔF-BAR)−6His (Δ2–279, pSP1940) and MBP-Hof1(ΔSID) −6His (Δ293–355[42], pSP1941) were generated by mutagenic PCR using primers MP1597-MP1598 and MP1599-MP1600, respectively. Gibson assembly (New England Biolabs) was used to construct plasmids for the expression of MBP-Hof1-P4A-6His (pSP1784), MBP-Hof1-P4E-6His (pSP1785) and MBP-Hof1-BFP-6His (pSP1950). For phosphomutants, *HOF1* DNA fragments (coding for aa 257–587 of Hof1) containing the 10 substitutions of *HOF1-P4A* (S313A, S314A, S337A, T341A, T350A, S366A, S421A, S423A, S424A, S563A) or *HOF1-P4E* (S313E, S314E, S337E, T341E, T350E, S366E, S421E, S423E, S424E, S563E) were amplified from pSP1721 (pTwist bearing a *HOF1-P4A* fragment from 1254 bp of the ORF to 121 bp after the stop codon) or pSP1722 (pTwist bearing a *HOF1-P4E* fragment from 1254 bp of the ORF to 121 bp after the stop codon) using primers MP1257 and MP1218, and cloned into pBG2163 by Gibson assembly using primers MP1444 and MP1454. For Hof1-BFP, an *mTagBFP* DNA fragment was amplified from pSP1706 (pHPOM1-mTagBFP2) using primers MP1614 and MP1495, and cloned by Gibson assembly into pBG2163[54] that was amplified with primers MP1492 and MP1493 to generate pSP1950. The list of primers and plasmids used in this study can be found in Tables S2 and S3.

## Live cell imaging

For time-lapse video microscopy, cells were mounted on 1% agarose pads in SD medium on fluorodishes and filmed at a controlled temperature. For scoring septin ring splitting, cells were filmed with a 100 × 1.49 NA oil immersion objective mounted on a Nikon Eclipse Ti microscope equipped with an EMCCD Evolve-512 Camera (Photometrics) and iLAS[2] module (Roper Scientific) and controlled by Metamorph. Z-stacks of 8 planes were acquired every 4 min with a step size of 0.4 μm and a binning of 1. Z-stacks were max-projected with ImageJ or Metamorph. *p-values* in statistical analyses were calculated using a two-sided $\chi^2$ test.

For quantifying fluorescence intensities associated to mCherry-Cdc3 and Hof1-eGFP and for resolving single from double Hof1 rings, cells were filmed with a 60× UPLXAPO 1.42 NA oil immersion objective mounted on an Olympus IX83 inverted microscope coupled to Yokogawa W1 spinning disk unit, equipped with a sCMOS Fusion BT camera (Hamamatsu) and controlled by the CellSens software. Z-stacks of 11–15 planes were acquired every 1–2 min with a step size of 0.24 μm, and max-projected using ImageJ. Alternatively (Figs. 3C, S6B), cells were filmed with a 100× Plan Apo lambda 1.45 NA oil immersion objective mounted on a Nikon inverted microscope coupled to the Andor Dragonfly spinning disk, equipped with an EMCCD iXon888 Life (Andor) camera and controlled by Andor fusion software. Focus was maintained using the Perfect Focus System (Nikon). Z-stacks of 8 planes were acquired every 1–2 min with a step size of 0.4 μm, and deconvolved (6 iterations, denoising filter size = 1 and pre-sharpening = 2). Z-stacks were max-projected using ImageJ.

For still images of septin collars in *hof1Δ* mutant (Fig. 1C), live cells were washed with PBS, mounted on slides with coverslips and immediately imaged at room temperature with a 100 × 1.4 NA Plan Apochromat oil immersion objective mounted on an epifluorescent widefield Zeiss Axioimager Z2 microscope equipped with an sCMOS ZYLA (Andor) camera and controlled by the Zen software. Z-stacks of 15 planes were acquired with a step size of 0.3 μm and max-projected with ImageJ.

## Quantification of septin- and Hof1-associated fluorescence intensities in yeast

Fluorescence intensities associated to mCherry-Cdc3 and Hof1-eGFP were quantified with ImageJ on max-projected Z-stacks.

Intensities were measured within a rectangular ROI defining the bud neck. A rolling ball radius of 50 pixels was set to subtract the background and fluorescent intensities were determined using ImageJ Analyze Particles tool after applying a threshold. The highest fluorescence integrated intensity reached at any time point within the time frame of interest was set at 100%, while all the other fluorescence intensities were relative to this reference value. Relative intensities were then averaged after setting as time 0 the time immediately preceding septin ring splitting. The number of samples analysed (*n*) is indicated in the figures. Error bars correspond to standard deviations.

## Protein extracts, immunoprecipitations, lambda phosphatase assays and western blotting

For TCA protein extracts, 10–15 ml of cell culture in logarithmic phase ($OD_{600}$ = 0.5–1) were collected by centrifugation at 2000 × *g*, washed with 1 ml of 20% TCA and resuspended in 100 μl of 20% TCA before breakage of cells with glass beads (diameter 0.5–0.75 mm) on a Vibrax VXR (IKA). After the addition of 400 μl of 5% TCA, lysates were centrifuged for 10 min at 845 × *g*. Protein precipitates were resuspended in 100 μl of 3× SDS sample buffer (240 mM Tris-Cl pH 6.8, 6% SDS, 30% glycerol, 2.28 M β-mercaptoethanol, 0.06% bromophenol blue), denatured at 99 °C for 3 min, and loaded on SDS−PAGE after elimination of cellular debris by centrifugation (5 min at 20,000 × *g*).

For coimmunoprecipitations, cells were lysed in TBSN buffer (100 mM NaCl, 25 mM Tris-Cl pH 7.4, 2 mM EDTA, 0.1% NP-40, 1 mM DTT), supplemented with 1 mM sodium orthovanadate, 60 mM β-glycerophosphate and a cocktail of protease inhibitors (Complete EDTA-free; Roche) at 4 °C with 14 cycles of 30″ breakage followed by 30″ in ice. 10 ODs of cleared extracts were diluted in 300 μl of TBSN supplemented with inhibitors, out of which 5 μl were withdrawn and denatured in 3× SDS sample buffer for inputs. The remaining IP mix was incubated for 1 h at 4 °C on a nutator with 25 μl of protein G dynabeads (Invitrogen) preadsorbed for 90 min with 1 μl of anti-Flag M2 antibody and washed with TBSN to eliminate the excess of antibody. Magnetic beads were washed six times with TBSN buffer and bound proteins eluted with an excess of 3×Flag peptide (0.5 mg/ml in 50 mM Tris-Cl pH 8.3, 1 mM EDTA, 0.1% SDS) upon incubation at R.T. for 25 min with vigorous shaking. 12.5 μl of 3× SDS sample buffer (240 mM Tris-Cl pH 6.8, 6% SDS, 30% glycerol, 2.28 M β-mercaptoethanol, 0.06% bromophenol blue) were added to the eluted proteins, followed by denaturation at 99 °C for 3 min and loading on SDS−PAGE.

For lambda phosphatase assays, Hof1-3HA was immunoprecipitated with the same procedure above, except that protein G dynabeads were coated with 12CA5 anti-HA antibody. After extensive washes, immunoprecipitates were incubated at 30 °C for 30 min in 30 μl of lambda phosphatase buffer containing 1 mM MnCl2 and 1 μl of lambda phosphatase (400 u/μl NEB Biolabs). 15 μl of 3× SDS sample buffer were added to the mix before denaturation at 99 °C for 3 min and loading on SDS−PAGE.

For western blotting, proteins were wet-transferred on Protran membranes (Schleicher and Schuell) overnight at 0.2 A. Specific proteins were detected with monoclonal anti-AID (M214-3 Medical and Biological Laboratories; 1:1000), anti-Pgk1 (22C5D8 Invitrogen Molecular Probes, 1:40,000) anti-GFP (3H9 Chromotek; 1:1000), anti-Pkc1 (sc-6801 Santa Cruz; 1:5000), anti-HA 12CA5 (AgroBio custom made; 1:5000), anti-Flag M2 (F3165 Sigma, 1:5000), anti-Clb2 (kind gift of E. Schwob) or polyclonal anti-Cdc11 (sc-7170 Santa Cruz; 1:2000). Antibodies were diluted in 5% low-fat milk (Regilait) dissolved in TBST (25 mM Tris-Cl pH 7.5, 137 mM NaCl, 2.68 mM KCl, 0.1% Tween-20). Secondary antibodies were purchased from GE Healthcare, and proteins were detected by a homemade luminol/p-coumaric acid enhanced chemiluminescence system.

## Flow-cytometric analysis of DNA contents

For DNA quantification by flow cytometry, $1-2 \times 10^7$ cells were collected, spun at $10,000 \times g$ and fixed with 1 ml of 70% ethanol for at least 30 min at RT. After one wash with 50 mM Tris-Cl pH 7.5, cells were resuspended in 0.5 ml of the same buffer containing 0.05 ml of a preboiled 10 mg/ml RNAse solution and incubated overnight at 37 °C. The next day cells were spun at $10,000 \times g$ and resuspended in 0.5 ml of 5 mg/ml pepsin freshly diluted in 55 mM HCl. After 30-min incubation at 37 °C, cells were washed with FACS buffer (200 mM Tris-Cl pH 7.5, 200 mM NaCl, 78 mM MgCl₂) and resuspended in the same buffer containing 50 µg/ml propidium iodide. After a short sonication, samples were diluted (1:8) in 200 µl of 50 mM Tris-Cl pH 7.5. 20,000 events were scored and analysed for each sample on an ACEA NovoCyte cytometer instrument controlled by NovoExpress software (Agilent).

## Protein purification

Yeast Cdc11-capped septin octamers were purified as previously described[76]. Briefly, *E.coli* strain BL21 (DE3) Rosetta cells were transformed with two bicistronic plasmid pFM453 (colE1, Amp^r, *CDC10, CDC11*) and pFM455 (p15A, Kan^r, *MBP-CDC12, His6-CDC3*) or with pFM453 and pFM873 (p15A, Kan^r, *MBP-CDC12, His6-CDC3-eGFP*)[77,99]. Cells were grown in LB containing 50 µg/ml of ampicillin, 25 µg/ml of kanamycin, 34 µg/ml of chloramphenicol, 0.2% glucose and induced at $OD_{600} = 1$ by addition of 0.2 mM IPTG. After 20 h at 16 °C, cells were harvested by centrifugation and resuspended in Buffer A (25 mM NaHPO₄ pH 7.8, 300 mM NaCl, 0.5 mM MgCl₂, 5% glycerol) supplemented with 5 mM β-mercaptoethanol, 1 mg/ml of lysozyme and a cocktail of protease inhibitors (Complete EDTA-free Roche). Cells were sonicated 3× with 1′30″ cycles of 8″ pulses/8″ ice and extracts cleared at $30,000 \times g$ for 30′ at 4 °C. Lysates were incubated 2 h at 4 °C with 1 ml of amylose resin (New England Biolabs), pre-washed 3× with buffer A, on a rotating wheel. After incubation with the protein extracts, the slurry was washed 3 times with buffer A and loaded on a Polyprep column (Bio-Rad). Fractions of 0.5 ml were eluted with buffer A supplemented with 10 mM maltose and quantified by Nanodrop. The most concentrated fractions were pulled together and MBP was cleaved from the complex using thrombin at 5 U/mg of protein overnight at 6 °C. Proteins were then diluted in 10 ml of buffer B (25 mM NaHPO₄ pH 7.8, 300 mM NaCl, 0.5 mM MgCl₂, 5% glycerol, 0.1% Triton X100, 6 mM imidazole) supplemented with 5 mM β-mercaptoethanol and incubated 2 h at 4 °C with 1 ml of Ni-NTA agarose beads (Qiagen), pre-washed 3x with buffer B, on a rotating wheel. The slurry was then washed 3 times with buffer C (25 mM NaHPO₄ pH 7.8, 500 mM NaCl, 0.5 mM MgCl₂, 5% glycerol, 0.15% Triton X100, 5 mM β-mercaptoethanol, 8 mM imidazole) and loaded on a Polyprep column (Bio-Rad). Fractions of 0.5 ml were eluted with buffer A supplemented with 200 mM imidazole and quantified by Nanodrop. The most concentrated fractions were dialysed in a buffer containing 20 mM Tris-Cl pH 8.2, 300 mM NaCl, 0.2 mM MgCl₂ and 2 mM DTT. Proteins were finally concentrated through Amicon Ultra filter units (10 kDa cut-off).

Recombinant Hof1 and mutant derivative proteins were purified as MBP fusions. For Hof1, Hof1-P4A, Hof1-P4E, Hof1-(ΔF-BAR), Hof1(Δ-SID) and Hof1-BFP purification, *E.coli* strain BL21 (DE3) Rosetta cells were transformed with pBG2163 (pMAL-C2 MBP-Hof1-6His[54]), pSP1784 (pMAL-C2 MBP-Hof1-P4A-6His), pSP1785 (pMAL-C2 MBP-Hof1-P4E-6His), pSP1940 (pMAL-C2 MBP-Hof1(ΔF-BAR)−6His), pSP1941 (pMAL-C2 MBP-Hof1(ΔF-SID)−6His) and pSP1950 (pMAL-C2 MBP-Hof1-BFP-6His). Cells were grown in LB containing 50 µg/ml of ampicillin, 34 µg/ml of chloramphenicol, and 0.2% glucose, to late log phase ($OD_{600} = 0.7-0.9$). Expression was induced with 0.8 mM IPTG overnight at 18 °C, and then cells were pelleted and stored at −80 °C. Pellets were thawed, and resuspended in lysis buffer (20 mM Tris-Cl, pH 8, 300 mM NaCl) supplemented with 1 mg/ml of lysozyme and a cocktail of protease inhibitors (Complete EDTA-free, Roche). Cells were sonicated 3× with 1′30″ cycles of 8″ pulses/8″ ice and extracts cleared at

$30,000 \times g$ for 30 min at 4 °C. Lysates were incubated 1 hr at 4 °C with 1.5 ml of Ni-NTA agarose beads (Qiagen), pre-washed 3× with 20 mM Tris-Cl pH 8, 300 mM NaCl, 6 mM imidazole at 4 °C on a rotating wheel. After incubation with the protein extracts, the slurry was washed 3 times with 20 mM Tris-Cl pH 8, 300 mM NaCl, 8 mM imidazole and loaded on a Polyprep column (Bio-Rad). Fractions of 0.5 ml were eluted with 20 mM Tris-Cl pH 8, 300 mM NaCl, 300 mM imidazole and quantified by Nanodrop. The most concentrated eluted fractions were pooled and diluted fourfold with lysis buffer lacking imidazole, mixed with 1.5 ml amylose beads and incubated at 4 °C for 1 h. The beads were washed five times with 10 ml lysis buffer and loaded on a Polyprep column (Bio-Rad). Fractions of 0.5 ml were eluted with lysis buffer supplemented with 20 mM maltose and 1 mM DTT and quantified by Nanodrop. The most concentrated fractions were dialysed in a buffer containing 50 mM Tris-Cl pH 7.5, 50 mM KCl, 150 mM NaCl, 1 mM EDTA, 1 mM DTT. Proteins were finally concentrated through Amicon Ultra filter units (50 kDa cut-off), snap frozen in liquid N₂ and stored at −80 °C.

## Dynamic light scattering

For dynamic light scattering, we used a Litesizer 500 setup (Anton Paar; scattering angle of 90° in-vacuo laser wavelength 658 nm, maximum power 40 mW). Recombinant septin octamers at 180 nM were polymerised in solution by lowering the salt concentration to 30 mM NaCl, and immediately measured. After 3 min, 180 nM of either recombinant Hof1 or Hof1-P4E were added. The hydrodynamic radius was obtained from a standard second-order cumulant fit.

## Transmission electron microscopy

Cdc11-capped septin octamers were diluted to 360 nM in low salt buffer (20 mM Tris-Cl pH 8, 30 mM NaCl, 2 mM MgCl₂, 40 µM GTP) and let polymerise for at least 30 min at 4 °C. Then, septins were mixed with recombinant Hof1 or Hof1-P4E (molar ratio 1:2, 1:1 or 1:0.5), diluted in the same buffer to 180 nM final concentration. 3 µl of samples were absorbed on glow-discharged formvar carbon-coated grids (Delta Microscopies FCF100-Cu) and stained with 1% uranyl acetate. Grids were observed using a JEOL 1400 Flash transmission electron microscope at 120 kV. Micrographs were recorded using a One View camera (Gatan Inc.) at different magnifications.

## Fluorescence microscopy and supported lipid bilayers

In all experiments, glass coverslips and slides were cleaned by three successive rounds of sonication for 30 min in 1 M NaOH, 100% ethanol and milliQ water, with extensive rinsing in double-distilled water between each step. Glass coverslips and slides were rinsed in 100% ethanol, dried and finally plasma cleaned just before using.

To visualise the formation of septin assemblies in solution, a glass coverslip was attached to a glass slide and several -20 µl flow chambers were made by placing strips of parafilm parallel to the shorter length of the slide followed by a short incubation of the slides at 100 °C to seal the edges. A mix of recombinant eGFP-tagged and untagged septin octamers (ratio 1:1) was diluted to 200 nM in polymerisation buffer (20 mM Tris-HCl pH 8, 30 mM NaCl, 2 mM MgCl₂, 40 mM GTP). Then, septins were mixed with 50% mol of recombinant Hof1 or Hof1-P4E in the same buffer to 50 nM final concentration. 20 µl of the sample was added to a flow chamber that was immediately sealed with modelling clay. Alternatively (Fig. S8B), 3 µl of the sample was added between a glass slide and coverslip and sealed with nail polish. Images were acquired at room temperature with a 100×1.4 NA Plan Apochromat oil immersion objective mounted on an epifluorescence widefield Zeiss Axioimager Z2 microscope equipped with an sCMOS ZYLA (Andor) camera and controlled by the Zen software. Z-stacks of 10 planes were acquired with a step size of 0.3 µm and max-projected with ImageJ. The background was subtracted and denoised using ImageJ tools. For quantification, fluorescence intensities associated to septins-eGFP

structures were quantified with ImageJ on max-projected Z-stacks. A rolling ball radius of 50 pixels was set to subtract the background and fluorescent intensities were measured using Analyze Particles tool after applying an appropriate threshold to highlight the structures of interest. The distribution of integrated densities was plotted and $p$-values calculated using an unpaired two-tailed $t$-test for statistical analyses.

Supported lipid bilayers were composed of either 95,8% (w/w) DOPC, 4% (w/w) DGS-NTA(Ni), 0,2% (w/w) Rhodamine-PE or 77% (w/w) Egg-PC, 2.8% (w/w) brain PI(4,5)$P_2$, 20% (w/w) liver PI and 0.2% (w/w) TopFluor-TMR-PI(4,5)P2. Lipids dissolved in methanol/chloroform (1:2) were mixed and dried for 1 h in a vacuum oven at 60 °C. Small unilamellar vesicles (SUVs, diameter ~100 nm) were obtained by extrusion of multilamellar vesicles in citrate buffer (citric acid - trisodium citrate 20 mM pH 4.6, KCl 50 mM, EGTA 0.5 mM) heated to 44 °C. 40 μl of SUVs were deposited on a two-well silicone insert (IBIDI), and incubated 1 h at 40 °C. Bilayers were carefully rinsed with washing buffer (20 mM Tris-Cl pH 8, 300 mM NaCl) and then equilibrated with equilibration buffer (20 mM Tris-Cl pH 8, 30 mM NaCl, 1 mg/ml lipid-free BSA). Cdc11-capped eGFP-tagged and untagged septin octamers were separately diluted in polymerisation buffer (20 mM Tris-Cl pH 8, 30 mM NaCl, 2 mM MgCl$_2$, 40 mM GTP), mixed in a 1:1 ratio (eGFP-tagged:untagged), and incubated for at least 30 min on ice. Purified Hof1 proteins were diluted in the same buffer. 50 nM of septins octamers in 20 μL were applied on the lipid bilayer and immediately visualised with a Zeiss LSM880 Airyscan confocal microscope. Then, 50, 100 or 250 nM of either Hof1 or Hof1-P4E in a volume of 20 μl were added. For Fig. S8A, Hof1-BFP-tagged and untagged mixed in a 1:4 ratio (tagged:untagged) were added in a volume of 20 μl. PI(4,5)$P_2$-containing SLBs were visualised with a Zeiss LSM880 Airyscan confocal microscope using an Argon laser for 488 nm and 514 nm as excitation sources. Acquisitions were performed with a 63×/1.4 NA objective. Multidimensional acquisitions were acquired via an Airyscan detector (32-channel GaAsP photomultiplier tube (PMT) array detector), controlled by Zeiss Zen software. PI(4,5)$P_2$-containing SLBs with Hof1-BFP and DOPC-containing SLB were visualised with a 60X UPLXAPO 1.42 NA oil immersion objective mounted on an Olympus IX83 inverted microscope coupled to Yokogawa W1 spinning disk unit, equipped with a sCMOS Fusion BT camera (Hamamatsu) and controlled by the CellSens software. Images were analysed by ImageJ and 3D projections were created with the Imaris software using the normal shading projection tool. For quantification, fluorescence associated to septins-GFP structures was quantified with ImageJ on 3D volumes of $20 \times 20 \times 6.5$ μm. A rolling ball radius of 50 pixels was set to subtract the background and fluorescent intensities were determined using the 3D object counter plugin after applying a threshold to highlight septin structures. The distribution of integrated densities was plotted. The $p$-values in statistical analyses were calculated using an unpaired two-tailed $t$-test.

## Statistics and reproducibility

Plots and statistical tests were performed using Excel or the GraphPad Prism software. Values from independent experiments are displayed as mean ± SD and the statistical test used is indicated in the figure legends. $P$-values are indicated in the figure legends. No statistical method was used to predetermine the sample size. No data were excluded from the analyses. The experiments were not randomised. The Investigators were not blinded to allocation during experiments and outcome assessment.

## Reporting summary

Further information on research design is available in the Nature Portfolio Reporting Summary linked to this article.

## Data availability

The authors declare that all data supporting the findings of this study are available within the paper and its supplementary information files. Source data are provided with the paper. Source data are provided with this paper.

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

## Acknowledgements

We are grateful to A. Amon, Y. Barral, M. Farkasovsky, M. Geymonat, B. Goode, M. Kanemaki, E. Schwob, D. Stillman, N. Talarek and R. Visintin for providing reagents; to C. Cazevieille at COMET electron microscopy facility for assistance with TEM specimen preparation; to F. El Alaoui for purification of fluorescent septin octamers; to M. Mavrakis for useful comments on the manuscript; to members of the Piatti and Liakopoulos teams for fruitful discussions. This work has been supported by a grant of the Agence Nationale pour la Recherche (SEPTORG ANR-18-CE13-0015-01) to S.P. and L.P., Fondation pour la Recherche Médicale (FRM project EQU202303016309) to S.P. and Fondation pour la Recherche Médicale (FRM project FDT202106012881) to M.V.S. I.A. was funded by a fellowship of the Labex EpiGenMed. We acknowledge the imaging facility MRI, member of the national infrastructure France-BioImaging supported by the French National Research Agency (ANR-10-INBS-04, «Investments for the future»).

## Author contributions

Conceptualisation: S.P. and M.V.S. Investigation: M.V.S, I.A., S.T., S.I., L.C. and S.P. Methodology: M.V.S, I.A., S.I., J.L.-K.-H., A.A., L.C., L.P. and S.P. Supervision: S.P. and L.P. Funding acquisition: S.P. and L.P. Writing: S.P. with inputs from authors.

## Competing interests

The authors declare no competing interests.
