## [Peer Review File · Nature Communications]

Phosphorylation of the F-BAR protein Hof1 drives septin ring splitting in budding yeastReviewer #1 (Remarks to the Author):

This manuscript describes a very careful and clearly written study of the effects of Hof1 phosphorylation on septin ring splitting. The Piatti group has done some very nice previous work setting the stage for this analysis, and the current work is similarly thorough and incisive. The data are for the most part very clear, and the interpretation is appropriate. The advance in understanding is significant, and this study will benefit the field. I have a few suggestions for hopefully simple ways to improve the work to provide further mechanistic insights and firm up the authors' model.

Major comments:

(1) Is it customary for phosphomimetic mutants to migrate more slowly on SDS-PAGE? In the Discussion, the authors imply that the mobility shift represents a "critical threshold of negative charges", and that makes some sense to me, but I haven't seen that concept demonstrated before in the literature. From what I understand, in standard SDS-PAGE, charge can have rather unpredictable effects on electrophoretic mobility, considering that in principle the entire protein should be totally coated in SDS and therefore highly negatively charged regardless of a few substitutions. Is there a published precedent for this "critical threshold" idea? If so, it would be worth citing. An alternative hypothesis is that the shift results from new phosphorylation on other residues as an indirect result of the mutations. Treatment with phosphatase prior to SDS-PAGE would address this possibility.

(2) The experimental approach in Figure 6 has great potential to provide direct mechanistic insights with regard to the proposed model, but some important aspects of the model are not tested, and it seems that they could be tested with relatively minor variations:

Where is Hof1 acting to bring about these changes in septin filaments? Since Hof1 has affinity for membranes, is it acting at the septin-membrane interface, or away from the membrane? Can the authors fluorescently label the Hof1 and co-localize it?

Since this experiment involves membranes and possible effects of Hof1 on membranes via the F-BAR domain are an important part of the model, can they test the Δ F-BAR mutant (or, better, point mutants that disrupt membrane binding; see below)?

The effects of phosphorylation are addressed via phosphomimetic mutations; if Mob1-Dbf2 is the candidate kinase, could the authors add purified Mob1-Dbf2 and look for similar effects? Alternatively, would Hof1 purified from yeast cells synchronized at the appropriate stage of the cell cycle have similar effects as the phosphomimetic mutant, which are then altered by prior treatment with phosphatase?

(3) The F-BAR domain deletion is a rather blunt mutation, and as the authors point out, F-BAR domains may be involved in oligomerization rather than membrane deformation. Considering that the F-BAR domain of Hof1 seems pretty similar to, and the AlphaFold structure prediction looks very much like, well-characterized F-BAR domains that deform membranes, I think it could be very illuminating to attempt to specifically mutate conserved residues that, in other F-BAR domains, directly interact with membranes (see for example PMID: 18329367), and test both in vivo phenotypes and effects on the in vitro experiments.

Minor comments:

line 99-100: "Consistently, S.c. Hof1 and S.p.Cdc15 have been involved in polarised growth": perhaps a better term would be "have been implicated", or else "are involved".

line 119: "and in particular its downstream Cdc14 phosphatase": I often get confused when writing refers to "protein X phosphatase" or "protein Y kinase", because I don't know if the phosphatase is protein X, or if it is removing phosphates from protein X. I recommend rewording to "and in particular its downstream phosphatase, Cdc14, as a ..."

Line 190: "While the large majority of GAL1-DMA2s and HOF1-3xminiAID cells": I suggest replacing "and" with "or", to prevent confusion.

Line 211: "rising prior to cytokinesis": I suggest "arising prior to cytokinesis"

Line 243: "FACS analysis": was this actually fluorescence-activated cell sorting, or just flow cytometry?

In the paragraph introducing the results with the analogue-sensitive *cdc15-as1 CDC14 TAB6-1* mutant, I had a hard time following the logic of the experiment until the interpretation was given at the end of the paragraph. I recommend rewording to begin this paragraph with an experimental prediction testing a hypothesis, then introducing the approach/mutant used to test the prediction/hypothesis. I think this is a place where it is worth it to use the extra words to spell out the logic.

In Figure 2F and H, I initially assumed that the red color distinguished statistically significant differences from non-significant (blue), but then the figure legend indicates that the color distinguishes above 50% from below 50%. I think it would be worth adding ">50%" in red text and "<50%" in blue text to the figure next to the "legend" that indicates red/blue for "splitting" and gray for "no splitting".

Line 299: The use of "significantly" here makes it sound like the slight sensitivity of the other mutants is insignificant, which isn't fair to say from a statistical or a biological perspective. I would replace "significantly" here with "severely".

Figures 2A, 4E, 5A: It appears that some sort of curve fitting was used to draw the lines connecting the data points in these plots. I don't think that's appropriate here, and I recommend using only straight lines to connect the points, otherwise it implies that there is some other information used to extrapolate about what the values in between data points would be.

Figure 5 shows effects of purified Hof1 on septin bundling in vitro. I think the relationship between septin bundling and septin ring splitting is not very clear, particularly in the absence of membranes. Thus, while these data are interesting, I think they can be moved to a supplemental figure, since the data in Figure 6 are more relevant to septin ring splitting.

Line 422: Another interpretation of the observation that "phosphomimetic mutants do not cause premature Hof1 relocalisation to the AMR in unperturbed conditions" is that the mutation to glutamate does not fully mimic the effects of phosphorylation in this context.

Lines 405-431 represent a large, unbroken block of text. To improve readability, I recommend breaking up into two or three paragraphs using line breaks (no change in text necessary).

Line 433: "close conformation" should be "closed conformation"

Line 443: The authors bring up the idea of potential redundancy, and mention as candidates "proteins whose inactivation is synthetically lethal with *hof1* deletion (i.e. *Cyk3*, *Chs2*, *Rvs167*)". This is an excellent point; from a more mechanistic perspective, does any of these other proteins have an F-BAR domain or another way to deform membranes?

Line 455: "albeit" should be "although" here.

Line 457: "cell separation cells" typo.

Line 468: "the genotype of most strains has been corrected to carry full-length *BUD4*": I was wondering about the potential contribution of *Bud4*, and this statement concerns me a little. Could the authors note here specifically which strains retain the *bud4-G2459fs* allele?

Reviewer identity: Michael McMurray, PhD, University of Colorado Anschutz Medical Campus

Reviewer #2 (Remarks to the Author):

Key Results

This manuscript from Salgado and colleagues addresses how the phosphorylation state of Hof1 regulates septin ring splitting dynamics in budding yeast *S. cerevisiae*. Genetics and live imaging are used to show that kinases Cdc15 and Mob1-Dbf2 are critical targets for the Mitotic Exit Network (MEN) phosphatase Cdc14 in promoting splitting of the septin ring. This finding leads the group to then focus on Hof1, an F-BAR protein that is a key target of Mob1-Dbf2 and known to regulate certain aspects of cytokinesis. Using *hof1* Δ and auxin-inducible *hof1*-AID strains, the authors show that disrupting Hof1 leads to defects in septin ring splitting. Next, they integrate a series of phospho-site mutants in the intrinsically disordered region (IDR) of Hof1, designed based on previous mass spectrometry results. Phospho-mimetic alleles of *hof1* suppressed lethality in a MEN shutoff strain and rescued the septin ring splitting defects. Constitutive Hof1 phosphorylation led to decreased binding of Hof1 to septin subunit Cdc10, assessed by immunoprecipitation assays, and diminished localization to the septin rings during cytokinesis. The group then addressed the role of the F-BAR domain of Hof1 and found that combining a *hof1* Δ F-BAR and their phospho-mimetic yeast constructs lead to growth defects at elevated temperatures and disrupted rescue of septin ring splitting when the MEN pathway is shutoff. Lastly, they used biochemical reconstitution to show that phosphorylation of Hof1 disrupts its ability to bundle septin filaments.

Validity

The genetic data are rigorous. The use of deletion, auxin-inducible degradation, galactose-inducible shutoff and integrated point mutants encompasses a wide scope of genetic perturbations to test the role of Hof1 in septin ring splitting. The live cell imaging nicely displays quantified defects in septin ring splitting. The role of Hof1 phosphorylation in septin bundling is assessed using immunoprecipitation assays, negative stain electron microscopy (EM), and fluorescence microscopy. These assays clearly show that phosphorylation of Hof1 decreases its interactions with septins *in vivo* and *in vitro*.

However, it is surprising that the authors omit discussion (or citation) of a key paper in their Hof1-septin reconstitution section (Garabedian et al., 2020). This 2020 paper demonstrated that purified Hof1 bundles septin filaments *in vitro*. Further, the authors make claims about the septin bundling activity of Hof1 being independent of membrane, when the Garabedian study already showed this clearly using EM and TIRF microscopy.

Significance

This manuscript contains a number of findings that will be of interest to the cell biology and biochemistry communities, as it helps clarify the role of Hof1 in controlling septin dynamics. Additionally, phosphorylation of F-BAR proteins is emerging as a major *in vivo* regulatory mechanism in controlling transitions between interphase and mitosis. Lastly, the role of PTMs affecting cytoskeletal dynamics is of interest to understanding how signal transduction affects septin/actin/microtubule binding protein regulation.

Data and methodology

The quality and presentation of the cell biology and genetic data are strong. However, the biochemistry showing effects of Hof1 phosphorylation and septin bundling require more rigorous quantification. The experiments in Figures 5, 6 and Figure S6 appear to only be done only in a single trial, and without quantification. Further, while Hof1 appears to have an effect on septin architecture, these effects must be quantified in order to understand the extent to which Hof1 phosphorylation changes septin architecture.

Analytical approach

The authors do not include any statistical tests for the data shown in their graphs. This is a major drawback to interpreting their findings. For instance, their graphs scoring septin ring splitting do not contain error bars or statistical tests (e.g., student's t-test or one-way ANOVA). The immunoprecipitation assays showing Hof1-septin interactions in cell lysates also require statistical tests. Lastly, as mentioned above, the Hof1-septin reconstitution in Figures 5, 6 and Figure S6 contain no meaningful quantification. This makes it difficult to discern the differences between their samples.

Suggested improvements

- In the immunoblot in Figure 2C, the wildtype Hof1-eGFP contains a single band. The authors make note of how their phospho-mimetic strains decrease the mobility of the Hof1-eGFP band due to increased charge similar to phosphorylation. However, the authors do not comment why the phospho-alanine strains all show up as a doublet band, which is also characteristic of a phospho-species. Also, if the authors want to equate these bands with phospho-species, conducting an in-vitro kinase assay with Mob1-Dbf2 and Hof1-eGFP should result in a mobility shift in SDS-PAGE. This can then be additionally validated with a lambda-phosphatase treatment to collapse the band.
- In Figure 3B, the authors do not make a note of the increased Hof1-P1A relative to their control pull-down. As mentioned in "Analytical approach" section, statistical tests would be helpful to assess if this increase is significant.
- The authors state that different lipid compositions in their bilayers did not affect Hof1-mediated septin bundling and thus is independent of lipids. Is this not implied from the EM in figure 5 which was done in the absence of lipids? Also, Garabedien et al., 2020 found Hof1-mediated septin bundling and this was not cited in discussing Figure 5.
- The authors show the phospho-mimetic Hof1 does not bundle septins well compared to wildtype Hof1. However, there is little speculation in the text as to how mechanistically this could translate into splitting of the septin ring. More insight about this would be an asset to the Discussion.
- In the Discussion the authors state, "Additionally, it should be noticed that none of our HOF1 alanine mutant completely abolishes Hof1 phosphorylation, thus providing a simple explanation for their lack of phenotype". I think either the authors need to make a yeast strain expressing a HOF1 allele that does abolish all phosphorylation sites in the IDR or speculate more as to why they did not see any phenotypes.

Clarity and context

It would be helpful for the authors to include a small figure about the GAL1-DMA2m strain that is used in so many assays. An example of this strain in the presence or absence of galactose showing septin ring splitting would provide the reader with a baseline of how to interpret the differences amongst tested strains.

Reviewer #3 (Remarks to the Author):

Salgado et. al, have shown a role for F-BAR protein Hof1 in inducing septin splitting prior to cytokinesis in budding yeast. They provide data to support their hypothesis that Hof1-induced septin splitting depends on Hof1 phosphorylation and its subsequent translocation to the AM-ring. They show that Hof1 phosphomimetic alleles the septin splitting defect observed in Dma2 overexpressing cells. Using biochemical and cell biological assays, they show that phosphorylation of Hof1 reduces its affinity for septin filaments and hence, causes its translocation to the AM-ring. Further, they also show that purified Hof1 protein can bundle septins filaments in vitro whereas its phosphomimetic mutant does so rather inefficiently as compared to the WT protein. The data presented in this paper does provide some clues towards role of Hof1 in inducing septin splitting but is largely unconvincing and does not lead to a strong conclusion about the role of Hof1 in inducing rearrangements of the septin scaffold in *S. cerevisiae*.

Major comments:

1. Fig 1: The literature or the data regarding the time resolution of Hof1 moving to the Actomyosin ring with respect to septin splitting is missing. It would be crucial to determine the order of these events and the extent of their overlap with a reasonably good time resolution (since these are very rapid events) to invoke the hypothesis that "Hof1 movement causes septin splitting".
2. The authors should probe for the phosphorylation status of Hof1 upon Dma2 overexpression. Has it been shown previously? This is important to determine why Hof1(wt and P4A) proteins are unable to rescue the no septin splitting phenotype caused by Dma2 overexpression.
3. Fig 1: The data provided shows that septin scaffold persists and doesn't split even after cytokinesis. However, it's not clear that how long does the septin collar persist after Hof1 deletion/depletion. Does it eventually disassemble? The authors should include the data about disassembly/clearance of the septin collar that do not split after depletion of Hof1.

4. The septin disassembly has been linked with local PTMs at the division site such as SUMOylation. The authors do not discuss or explore the PTM landscape upon Hof1 depletion and its possible contribution to the observed phenotype of "no splitting".

5. Fig. 2: Hof1 protein levels across cell cycle stages for the different Hof1 alleles should be shown for a comparison alongside the cell-cycle dependent phosphorylation events. Also, Hof1 has previously shown to be degraded by Grr1 during late mitosis and G1 phase through its PEST motif (Blondel et. al, 2005). Since some of the phosphorylation sites studied here lie in or near the PEST motif, the authors should comment on its significance with respect to their observations.

6. Fig.3: The localization of Hof1 proteins (WT, 4A, 4E) has not been shown when Dma2 is overexpressed. Does it change as compared to what is shown in Fig.3C?

7. Fig. 4: Fig.4B suggests that deletion of the F-BAR domain is important for Hof1 degradation and causes an increase in Hof1 levels. Could the increased Hof1 levels be the reason why Hof1- Δ F-BAR phospho-mimetic alleles do not show a rescue of phenotype as represented in Fig. 4D.

8. Fig. 5: The authors have performed in-vitro analysis of Hof1 effect on septin filaments reconstituted from purified septins. It would be more informative to add how the Hof1-P4A, Δ F-BAR and Hof1- Δ FBAR-phosphomimetic(E) mutant protein affect septin filament bundling. Hof1- Δ SID (septin-interacting domain) should also tested in this experiment.

9. In lines 292-296, the authors hypothesize that F-BAR domain of Hof1 may modify the plasma membrane at the AMR after translocation. But the question arises that the F-BAR domain is also present in the Hof1 WT and P4A proteins and these variants also show a translocation to the medial AMR, then why are these variants of Hof1 unable to rescue the no-splitting defect? This should be discussed and clarified.

Minor comments:

1. Fig1-B: Does the chain phenotype exhibiting no septin splitting represent the population of cell with the cell cycle arrest or the delay in cell cycle progression due to inactivated MEN signalling?

2. Authors should represent the Fig1-E data prior to Fig-D. And both the panels should indicate whether the inducer is IAA or NAA for Hof1-3xminiAID.

3. Authors have not specified the type of inducer (IAA or NAA) used in Fig1-F and Fig1-G or in the text. Authors should represent the data with NAA in all these panels.

Reviewer #1

This manuscript describes a very careful and clearly written study of the effects of Hof1 phosphorylation on septin ring splitting. The Piatti group has done some very nice previous work setting the stage for this analysis, and the current work is similarly thorough and incisive. The data are for the most part very clear, and the interpretation is appropriate. The advance in understanding is significant, and this study will benefit the field. I have a few suggestions for hopefully simple ways to improve the work to provide further mechanistic insights and firm up the authors' model.

Major comments:

(1) Is it customary for phosphomimetic mutants to migrate more slowly on SDS-PAGE? In the Discussion, the authors imply that the mobility shift represents a "critical threshold of negative charges", and that makes some sense to me, but I haven't seen that concept demonstrated before in the literature. From what I understand, in standard SDS-PAGE, charge can have rather unpredictable effects on electrophoretic mobility, considering that in principle the entire protein should be totally coated in SDS and therefore highly negatively charged regardless of a few substitutions. Is there a published precedent for this "critical threshold" idea? If so, it would be worth citing. An alternative hypothesis is that the shift results from new phosphorylation on other residues as an indirect result of the mutations. Treatment with phosphatase prior to SDS-PAGE would address this possibility.

The Reviewer is totally right, amino acid charge has an unpredictable effect on proteins electrophoretic mobility and there is no formal demonstration that a critical threshold of negative charges causes an electrophoretic mobility shift. For instance, the yeast kinetochore protein Ask1 is phosphorylated by multiple mitotic kinases and undergoes a phosphorylation-dependent mobility shift during S and M phase. However, a single CDK phosphorylation site was found to be mainly responsible for the mobility shift (Li and Elledge, 2003, *Cell Cycle* 2:143). We have amended the misleading sentence in the discussion. For what concerns the second part of the Reviewer's question, we have done the suggested experiment and treated with lambda phosphatase Hof1-3HA immunoprecipitated from yeast cells. Results show that the protein mobility shift of the Hof1-P4E mutant protein does not collapse upon phosphatase treatment (Fig. S5A), suggesting that it is not due to new phosphorylations. Consistently, MBP-Hof1-P4E purified from bacterial cells show a similar mobility shift relative to MBP-Hof1 (Fig. R1 for Reviewers only).

(2) The experimental approach in Figure 6 has great potential to provide direct mechanistic insights with regard to the proposed model, but some important aspects of the model are not tested, and it seems that they could be tested with relatively minor variations:

Where is Hof1 acting to bring about these changes in septin filaments? Since Hof1 has affinity for membranes, is it acting at the septin-membrane interface, or away from the membrane? Can the authors fluorescently label the Hof1 and co-localize it?

This is a very good question for which at the moment we only have a partial answer. In our *in vitro* assays Hof1 induces septin bundling mainly in solution, above the membrane. We now show in the new Fig. S8A that BFP-tagged Hof1 is part of septin bundles. We envision that the septin-bundling activity of Hof1 contributes to the rigidity of the septin collar in mitosis. Shortly before septin ring splitting, Hof1 is delocalized from the septin collar to the AMR (new Fig. 1C). However, since *hof1* deletion/depletion mutants display septin splitting defects, we think that the dissociation of Hof1 from septins is not sufficient to cause septin

destabilization and rearrangement, but the translocation of Hof1 to the medial ring is equally important. Our finding that the F-BAR domain of Hof1 contributes to septin ring splitting is consistent with this conclusion. However, how it does so remains speculative at this stage and we propose different possible scenarios in the Discussion. Indeed, if and how Hof1 interacts with the membrane is an important question that will be addressed in future studies, especially since expression of Hof1 F-BAR domain in cultured cells does not induce membrane tabulation, unlike other F-BAR domains (Moravcevic et al., 2015). Unfortunately, our recombinant Hof1-BFP protein is not ideally suited to study Hof1 interaction with the membrane, because the BFP tag interferes with septin bundling and we cannot exclude that it also affects other properties of Hof1. For this reason, to study its colocalization with septin bundles (Fig. S8A) we had to mix it with untagged Hof1 at a ratio 1:4 (tagged:untagged). Considering that BFP is not a bright fluorophore, studying Hof1 interaction with membranes will require further optimization. It should also be noticed that the requirement of the F-BAR domain for Hof1 function in septin ring splitting could as well involve Hof1 oligomerization or interaction with other proteins, and we are going to explore these possibilities in the future. We are also currently thinking about ways to study septin reorganization and recapitulate septin ring splitting using *in vitro* assays on SLBs. Hopefully, in the future we will be able to fully address this point.

Since this experiment involves membranes and possible effects of Hof1 on membranes via the F-BAR domain are an important part of the model, can they test the Δ F-BAR mutant (or, better, point mutants that disrupt membrane binding; see below)?

For the moment we have tested only the F-BAR deletion mutant in our *in vitro* assays because it was already available in our lab, while testing *in vitro* the point mutants that we have generated *in vivo* (see below) would have considerably delayed the resubmission of our manuscript. Our experiments (new Fig. S8B) show that deleting the F-BAR domain severely compromises Hof1 septin-bundling activity. This is likely due to poor binding to septins, since the F-BAR domain of Hof1 was previously found to contribute to septin interaction (Meitinger et al., 2011).

The effects of phosphorylation are address via phosphomimetic mutations; if Mob1-Dbf2 is the candidate kinase, could the authors add purified Mob1-Dbf2 and look for similar effects? Alternatively, would Hof1 purified from yeast cells synchronized at the appropriate stage of the cell cycle have similar effects as the phosphomimetic mutant, which are then altered by prior treatment with phosphatase?

These experiments are quite challenging, partly because Mob1-Dbf2 is only active in the presence of the upstream MEN kinase Cdc15 (König et al., 2010). Additionally, some phosphorylation sites that are mutated in our *HOF1-P1* and *HOF1-P4* mutants are likely to be phosphorylated by kinases other than Mob1-Dbf2 and yet to be identified, which complicates the design of straightforward experimental set-ups. Nevertheless, we have tried to address this point by phosphorylating Hof1 *in vitro* through a hyperactive Mob1-Dbf2 kinase that becomes partially independent of Mob1 phosphorylation by Cdc15. Constructs for yeast overexpression and purification of GST-Mob1-Dbf2 (wild type, hyperactive or kinase-dead) were kindly provided by Dr. M. Geymonat (University of Cambridge) and had been already used by the Pereira's lab (Meitinger et al., 2011). Using radioactive assays, we found that Hof1 could be efficiently phosphorylated by hyperactive Mob1-Dbf2, but not by the wild type or the kinase-dead kinase (Fig. R1 for Reviewers only), confirming and extending previous data (Meitinger et al., 2011). Additionally, we found that much of the *in vitro* phosphorylation

was impaired by the *-4PA* and *-4PE* mutations (compare the phosphorylation of MBP-Hof1 proteins with MBP alone in Fig. R1), thus confirming that most serines and threonines phosphorylated by Mob1-Dbf2 are mutated in the *HOF1-P4* mutants. Unfortunately, Hof1 phosphorylation *in vitro* by Mob1-Dbf2 did not lead to any electrophoretic mobility shift even at very high ATP concentrations (e.g. 1-5mM, data not shown), which precluded any assessment of the fraction of Hof1 that was actually phosphorylated (Figure R1 for Reviewers only). When we tried to study the effects of Hof1 phosphorylated *in vitro* by Mob1-Dbf2 we found, surprisingly, that addition of GST-Mob1-Dbf2 alone to septins (i.e. in the absence of Hof1) caused prominent septin bundling (data not shown). The reason for this is unclear at the moment, but could be due to direct septin phosphorylation by Mob1-Dbf2 or to Hof1 co-purifying with GST-Mob1-Dbf2 (note that Mob1 and Hof1 physically interact; Meitinger et al., 2011). Thus, the results of these experiments were not conclusive. For what concerns Hof1 purification from yeast cells, we do not think that we would be able to purify enough material to perform the *in vitro* bundling assays. In addition, this kind of experiments would have the caveat of possible indirect effects by Hof1 interactors.

(3) The F-BAR domain deletion is a rather blunt mutation, and as the authors point out, F-BAR domains may be involved in oligomerization rather than membrane deformation. Considering that the F-BAR domain of Hof1 seems pretty similar to, and the AlphaFold structure prediction looks very much like, well-characterized F-BAR domains that deform membranes, I think it could be very illuminating to attempt to specifically mutate conserved residues that, in other F-BAR domains, directly interact with membranes (see for example PMID: 18329367), and test both in vivo phenotypes and effects on the in vitro experiments.

We have now generated by CRISPR/Cas9 three novel *HOF1* point mutants where basic residues in the concave surface of the F-BAR domain have been changed into glutamate, to introduce opposite charges and interfere with membrane binding. These mutants have been designed according to the crystal structure of the Hof1 F-BAR domain (Lemmon et al., 2014) and may contribute to binding to phosphoinositides according to previous studies on other F-BAR proteins (Tsujita et al., 2006; Shimada et al., 2007). This approach has been largely exploited to study the interaction of F-BAR proteins with membranes and their ability to induce tubulation (e.g. Frost et al., 2008; Reider et al., 2009). Importantly, we find that these mutations impair the ability of Hof1-P4E to induce septin reorganization/clearance in *GALI-DMA2* overexpressing cells, thus recapitulating the effects of the F-BAR deletion. These data are shown in the new Fig. 4E-G.

Minor comments:

line 99-100: “Consistently, S.c. Hof1 and S.p.Cdc15 have been involved in polarised growth”: perhaps a better term would be “have been implicated”, or else “are involved”.

Done

line 119: “and in particular its downstream Cdc14 phosphatase”: I often get confused when writing refers to “protein X phosphatase” or “protein Y kinase”, because I don’t know if the phosphatase is protein X, or if it is removing phosphates from protein X. I recommend rewording to “and in particular its downstream phosphatase, Cdc14, as a ...”

Done

Line 190: “While the large majority of GAL1-DMA2s and HOF1-3xminiAID cells”: I suggest replacing “and” with “or”, to prevent confusion.

Done

Line 211: “rising prior to cytokinesis”: I suggest “arising prior to cytokinesis”

Done

Line 243: “FACS analysis”: was this actually fluorescence-activated cell sorting, or just flow cytometry?

It is indeed flow cytometry. We have corrected this throughout the text.

In the paragraph introducing the results with the analogue-sensitive cdc15-as1 CDC14 TAB6-1 mutant, I had a hard time following the logic of the experiment until the interpretation was given at the end of the paragraph. I recommend rewording to begin this paragraph with an experimental prediction testing a hypothesis, then introducing the approach/mutant used to test the prediction/hypothesis. I think this is a place where it is worth it to use the extra words to spell out the logic.

We rephrased the beginning of this paragraph as suggested.

In Figure 2F and H, I initially assumed that the red color distinguished statistically significant differences from non-significant (blue), but then the figure legend indicates that the color distinguishes above 50% from below 50%. I think it would be worth adding “>50%” in red text and “<50%” in blue text to the figure next to the “legend” that indicates red/blue for “splitting” and gray for “no splitting”.

We have corrected the legend of this figure (we could not find the < or > symbols in the Inkscape software that we used to make Figures; thus we replaced them by ≤ or ≥)

Line 299: The use of “significantly” here makes it sound like the slight sensitivity of the other mutants is insignificant, which isn’t fair to say from a statistical or a biological perspective. I would replace “significantly” here with “severely”.

Done

Figures 2A, 4E, 5A: It appears that some sort of curve fitting was used to draw the lines connecting the data points in these plots. I don’t think that’s appropriate here, and I recommend using only straight lines to connect the points, otherwise it implies that there is some other information used to extrapolate about what the values in between data points would be.

We have removed the curve smoothing from Figs. 2A, 3E and 5A.

Figure 5 shows effects of purified Hof1 on septin bundling in vitro. I think the relationship between septin bundling and septin ring splitting is not very clear, particularly in the absence of membranes. Thus, while these data are interesting, I think they can be moved to a supplemental figure, since the data in Figure 6 are more relevant to septin ring splitting.

As we explained above, Hof1 induces septin bundling both in solution and on membranes and we think that this activity may be relevant for the stability of the septin collar, rather than for septin ring splitting. Indeed, the data seem to indicate that Hof1 is no longer bound to septins when it promotes septin ring splitting. We would like to keep Figure 5 as a main figure because it shows the ultrastructure of septin assemblies and is complementary to the data in Figure 6.

Line 422: Another interpretation of the observation that “phosphomimetic mutants do not cause premature Hof1 relocalisation to the AMR in unperturbed conditions” is that the mutation to glutamate does not fully mimic the effects of phosphorylation in this context.

We agree, we have slightly modified the discussion accordingly.

Lines 405-431 represent a large, unbroken block of text. To improve readability, I recommend breaking up into two or three paragraphs using line breaks (no change in text necessary).

Done

Line 433: “close conformation” should be “closed conformation”

Done

Line 443: The authors bring up the idea of potential redundancy, and mention as candidates “proteins whose inactivation is synthetically lethal with hof1 deletion (i.e. Cyk3, Chs2, Rvs167)”. This is an excellent point; from a more mechanistic perspective, does any of these other proteins have an F-BAR domain or another way to deform membranes?

Rvs167 is an endocytic protein that carries an N-BAR domain able to induce membrane tabulation *in vitro*. If and how Cyk3 and Chs2 directly interact with the plasma membrane to our knowledge is not known. Future work in our lab will explore in detail these genetic interactions to try to identify other players that cooperate with Hof1 in septin ring splitting.

Line 455: “albeit” should be “although” here.

Corrected

Line 457: “cell separation cells” typo.

Corrected

Line 468: “the genotype of most strains has been corrected to carry full-length BUD4”: I was wondering about the potential contribution of Bud4, and this statement concerns me a little. Could the authors note here specifically which strains retain the bud4-G2459fs allele?

Almost all experiments have been performed with the corrected *BUD4*⁺ strains, except those in Figs. S1A-C, S5C, as we now state on page 21 of the manuscript. These data concern the initial characterization of *hof1*Δ mutant cells and the initial characterization of the viability of *HOF1* phosphorylation mutants in otherwise wild type cells at different temperatures.

Reviewer #2

Key Results

*This manuscript from Salgado and colleagues addresses how the phosphorylation state of Hof1 regulates septin ring splitting dynamics in budding yeast *S. cerevisiae*. Genetics and live imaging are used to show that kinases Cdc15 and Mob1-Dbf2 are critical targets for the Mitotic Exit Network (MEN) phosphatase Cdc14 in promoting splitting of the septin ring. This finding leads the group to then focus on Hof1, an F-BAR protein that is a key target of Mob1-Dbf2 and known to regulate certain aspects of cytokinesis. Using *hof1*Δ and auxin-inducible *hof1-AID* strains, the authors show that disrupting Hof1 leads to defects in septin ring splitting. Next, they integrate a series of phospho-site mutants in the intrinsically disordered region (IDR) of Hof1, designed based on previous mass spectrometry results. Phospho-mimetic alleles of *hof1* suppressed lethality in a MEN shutoff strain and rescued the septin ring splitting defects. Constitutive Hof1 phosphorylation led to decreased binding of Hof1 to septin subunit Cdc10, assessed by immunoprecipitation assays, and diminished localization to the septin rings during cytokinesis. The group then addressed the role of the F-BAR domain of Hof1 and found that combining a *hof1*ΔF-BAR and their phospho-mimetic yeast constructs lead to growth defects at elevated temperatures and disrupted rescue of septin ring splitting when the MEN pathway is shutoff. Lastly, they used biochemical reconstitution to show that phosphorylation of Hof1 disrupts its ability to bundle septin filaments.*

Validity

The genetic data are rigorous. The use of deletion, auxin-inducible degradation, galactose-inducible shutoff and integrated point mutants encompasses a wide scope of genetic perturbations to test the role of Hof1 in septin ring splitting. The live cell imaging nicely displays quantified defects in septin ring splitting. The role of Hof1 phosphorylation in septin bundling is assessed using immunoprecipitation assays, negative stain electron microscopy (EM), and fluorescence microscopy. These assays clearly show that phosphorylation of Hof1 decreases its interactions with septins in vivo and in vitro.

However, it is surprising that the authors omit discussion (or citation) of a key paper in their Hof1-septin reconstitution section (Garabedian et al., 2020). This 2020 paper demonstrated that purified Hof1 bundles septin filaments in vitro. Further, the authors make claims about the septin bundling activity of Hof1 being independent of membrane, when the Garabedian study already showed this clearly using EM and TIRF microscopy.

We apologize for this omission. We have now clearly referred to this paper in the Results section as soon as we talk about septin bundling by Hof1.

Significance

This manuscript contains a number of findings that will be of interest to the cell biology and biochemistry communities, as it helps clarify the role of Hof1 in controlling septin dynamics. Additionally, phosphorylation of F-BAR proteins is emerging as a major in vivo regulatory mechanism in controlling transitions between interphase and mitosis. Lastly, the role of PTMs affecting cytoskeletal dynamics is of interest to understanding how signal transduction affects septin/actin/microtubule binding protein regulation.

Data and methodology

The quality and presentation of the cell biology and genetic data are strong. However, the biochemistry showing effects of Hof1 phosphorylation and septin bundling require more rigorous quantification. The experiments in Figures 5, 6 and Figure S6 appear to only be done only in a single trial, and without quantification. Further, while Hof1 appears to have an effect on septin architecture, these effects must be quantified in order to understand the extent to which Hof1 phosphorylation changes septin architecture.

The experiments in these Figures were done multiple times, with similar results, although in the first version of the manuscript there was no quantification of the data. We have now quantified the fluorescence intensity associated to septin assemblies in the *in vitro* assays using fluorescent septins (new Fig. 6C). In contrast, TEM results (Fig. 5) cannot be quantified and are mainly meant to provide information about the ultrastructure of septin assemblies. To overcome this drawback, we have used dynamic light scattering (Fig. 5A) that provides quantitative data about the size of septin structures.

Analytical approach

The authors do not include any statistical tests for the data shown in their graphs. This is a major drawback to interpreting their findings. For instance, their graphs scoring septin ring splitting do not contain error bars or statistical tests (e.g., student's t-test or one-way ANOVA). The immunoprecipitation assays showing Hof1-septin interactions in cell lysates also require statistical tests. Lastly, as mentioned above, the Hof1-septin reconstitution in Figures 5, 6 and Figure S6 contain no meaningful quantification. This makes it difficult to discern the differences between their samples.

We have now added statistical tests to most of the relevant experiments (Fig. 1B, 2F, 3B, 4G, 6C, S8B), including scoring of septin ring splitting, immunoprecipitations for Hof1-septin interaction and *in vitro* septin bundling assays using GFP-tagged septins.

Suggested improvements

- In the immunoblot in Figure 2C, the wildtype Hof1-eGFP contains a single band. The authors make note of how their phospho-mimetic strains decrease the mobility of the Hof1-eGFP band due to increased charge similar to phosphorylation. However, the authors do not comment why the phospho-alanine strains all show up as a doublet band, which is also characteristic of a phospho-species. Also, if the authors want to equate these bands with phospho-species, conducting an in-vitro kinase assay with Mob1-Dbf2 and Hof1-eGFP should result in a mobility shift in SDS-PAGE. This can then be additionally validated with a lambda-phosphatase treatment to collapse the band.*

We have now carried out an extensive analysis of Hof1 phosphorylation during the cell cycle, as assessed by electrophoretic mobility shift, which was previously shown to collapse upon lambda phosphatase treatment (Meitinger et al., 2011). Our data (new Fig. S4) show that Hof1 phosphorylation is maximal at cytokinesis (t= 105' after release from G1 arrest). While phosphorylation is clearly reduced in *dbf20Δ dbf2-2* double mutants (new Fig. S4C), it is only mildly affected in *HOF1-P1A* and *HOF1-P4A* (Fig. S4A-B; compare the ratio between phosphorylated and unphosphorylated isoforms in wild type versus *HOF1-P1A/-P4A* cells at 90-105'), suggesting that not all Dbf2-dependent phosphorylation sites have been mutated in these mutants, as expected since all Dbf2-dependent phosphorylation sites are those mutated in the *HOF1-P2* mutants. These results are also consistent with Hof1 being phosphorylated by multiple kinases, including CDKs and polo kinase. However, it is also possible that in the absence of the main Dbf2-dependent phosphorylation sites secondary phosphorylations

appear on residues that are normally poorly phosphorylated, as previously shown for other phosphorylated proteins. In contrast, the phospho-mimetic *HOF1-P1E* and *HOF1-P4E* mutants cause a constitutive mobility shift of the Hof1 fast mobility isoform (Fig. S4A-B) that is not collapsed by lambda phosphatase treatment (Fig. S5A). Additionally, the Hof1-P1E and Hof1-P4E proteins still show cell cycle-dependent phosphorylation, which is again consistent with multiple kinases or secondary phosphorylation sites being responsible for Hof1 phosphorylation.

For what concerns the second part of this point, we have now carried out *in vitro* phosphorylation assays using recombinant MBP-Hof1 and GST-Mob1-Dbf2 complexes purified from yeast (see also our answer to Reviewer #1). In radioactive assays, we can clearly see that Mob1-Dbf2 can phosphorylate Hof1, as previously shown (Meitinger et al., 2011). These results are shown in Fig. R1 for Reviewers only. However, this phosphorylation does not cause any obvious mobility shift, even in the presence of high ATP concentrations (1-5 mM, Fig. R1 and data not shown). Thus, Hof1 phosphorylation appears to be very complex and will need to be further dissected in the future.

In Figure 3B, the authors do not make a note of the increased Hof1-P1A relative to their control pull-down. As mentioned in "Analytical approach" section, statistical tests would be helpful to assess if this increase is significant.

The statistical test has been added. The difference between Hof1 and Hof1-P1A was not significant.

The authors state that different lipid compositions in their bilayers did not affect Hof1-mediated septin bundling and thus is independent of lipids. Is this not implied from the EM in figure 5 which was done in the absence of lipids? Also, Garabedian et al., 2020 found Hof1-mediated septin bundling and this was not cited in discussing Figure 5.

The Reviewer is totally right, Hof1 induces septin bundling independently of membranes and membrane composition. As mentioned above, we have now cited the Garabedian paper in the Results while presenting Fig. 5.

The authors show the phospho-mimetic Hof1 does not bundle septins well compared to wildtype Hof1. However, there is little speculation in the text as to how mechanistically this could translate into splitting of the septin ring. More insight about this would be an asset to the Discussion.

Our model, which we have tried to better explain in the Discussion, envisions that Hof1 must leave septins and join the AMR at the medial ring to promote septin ring splitting. Hof1 dissociation from septin, which is triggered by its phosphorylation, is likely insufficient for Hof1 to bring about septin reorganization. Rather, its concomitant relocation to the medial ring may be instrumental for this process. For the moment, besides having implicated the F-BAR domain in this process, which suggests a membrane-bound function, we have little idea of the molecular mechanism underlying this process, although we propose different possible scenarios in the Discussion. Our future work will be devoted to unravel this fascinating, yet mysterious process.

In the Discussion the authors state, "Additionally, it should be noticed that none of our HOF1 alanine mutant completely abolishes Hof1 phosphorylation, thus providing a simple explanation for their lack of phenotype". I think either the authors need to make a yeast strain

expressing a HOF1 allele that does abolish all phosphorylation sites in the IDR or speculate more as to why they did not see any phenotypes.

As mentioned above, Hof1 phosphorylation is very complex. In *HOF1-P1A* we have mutated all the phosphorylation sites that we found to be regulated during mitotic progression. Other phosphorylations, which are constant throughout the cell cycle or appear under various conditions, may have a very different meaning and mutagenizing them all together would not necessarily help us understanding how Hof1 is regulated for its cytokinetic functions. A better approach would probably be to submit the Hof1-P1A and Hof1-P4A proteins to multiple rounds of mass spectrometry analysis and mutagenesis to study possible secondary phosphorylation sites. However, this is a very time-consuming procedure. We have now further speculated in the Discussion about the reasons why our alanine mutants do not show any obvious phenotype.

Clarity and context

It would be helpful for the authors to include a small figure about the GAL1-DMA2m strain that is used in so many assays. An example of this strain in the presence or absence of galactose showing septin ring splitting would provide the reader with a baseline of how to interpret the differences amongst tested strains.

Unfortunately, our figures are already very packed and we could not find a suitable place where to introduce this additional figure. However, the phenotype of *GAL1-DMA2m* cells in the presence of galactose, which is the most relevant for the interpretation of our results, is shown in Fig. 2D-E,G and was carefully characterized in previous papers from our lab (Fraschini et al., 2004; Tamborrini et al., 2018).

Reviewer #3

Salgado et. al, have shown a role for F-BAR protein Hof1 in inducing septin splitting prior to cytokinesis in budding yeast. They provide data to support their hypothesis that Hof1-induced septin splitting depends on Hof1 phosphorylation and its subsequent translocation to the AM-ring. They show that Hof1 phosphomimetic alleles the septin splitting defect observed in Dma2 overexpressing cells. Using biochemical and cell biological assays, they show that phosphorylation of Hof1 reduces its affinity for septin filaments and hence, causes its translocation to the AM-ring. Further, they also show that purified Hof1 protein can bundle septins filaments in vitro whereas its phosphomimetic mutant does so rather inefficiently as compared to the WT protein.

*The data presented in this paper does provide some clues towards role of Hof1 in inducing septin splitting but is largely unconvincing and does not lead to a strong conclusion about the role of Hof1 in inducing rearrangements of the septin scaffold in *S. cerevisiae*.*

Major comments:

1. Fig 1: The literature or the data regarding the time resolution of Hof1 moving to the Actomyosin ring with respect to septin splitting is missing. It would be crucial to determine the order of these events and the extent of their overlap with a reasonably good time resolution (since these are very rapid events) to invoke the hypothesis that “Hof1 movement causes septin splitting”.

We have done the suggested analysis of Hof1-eGFP localisation at the bud neck relative to septin ring splitting (visualized by mCherry-Cdc3). To increase the resolution of these structures we have used diploid cells and deconvolved the images after acquisition. The data,

shown in new Fig. 1C, show that Hof1 undergoes a visible rearrangement on average 2.5 minutes before septin ring splitting and completely collapses into a single contractile ring 0.3 minutes prior to septin ring splitting. This timing is in agreement with the idea that Hof1 contributes to septin reorganisation at cytokinesis.

2. The authors should probe for the phosphorylation status of Hof1 upon Dma2 overexpression. Has it been shown previously? This is important to determine why Hof1(wt and P4A) proteins are unable to rescue the no septin splitting phenotype caused by Dma2 overexpression.

We have now analysed the phosphorylation status of Hof1-3HA in synchronized *GAL1-DMA2m* cells progressing through the cell cycle. Unexpectedly however, Hof1-3HA was present at significantly higher levels in *GAL1-DMA2m* cells relative to the wild type control (new Fig. S7A), which precluded a meaningful analysis of its phosphorylation state. We have nevertheless decided to include these data in our manuscript for sake of completeness. The reason for this unexpected result is unclear at the moment and will be investigated in the future.

3. Fig 1: The data provided shows that septin scaffold persists and doesn't split even after cytokinesis. However, it's not clear that how long does the septin collar persist after Hof1 deletion/depletion. Does it eventually disassemble? The authors should include the data about disassembly/clearance of the septin collar that do not split after depletion of Hof1.

As discussed thoroughly at the beginning of the Results section, a careful characterization of the phenotype caused by *HOF1* deletion/depletion through live cell imaging is at present technically very challenging. Depletion experiments in YEPD using the AID system showed that in chained cells the oldest septin collars (which we can infer using alpha-factor arrested cells that make shmoos) tend to fade with time, possibly because of septin recycling. However, we do not feel comfortable to include these data in the paper, because in most cases it is very difficult to distinguish in still images chains of cells arising from cytokinesis defects from clumps of cells that are accidentally close to one another.

4. The septin disassembly has been linked with local PTMs at the division site such as SUMOylation. The authors do not discuss or explore the PTM landscape upon Hof1 depletion and its possible contribution to the observed phenotype of "no splitting".

Septin sumoylation was not linked to septin ring splitting, as a mutant where the sumoylated lysines of three septins were mutated to arginine did not show any septin splitting or cytokinesis defects, while it displayed a delay in septin disassembly after cytokinesis. Consistently, septin sumoylation is maximal during mitosis (Johnson and Blobel, 1999). There are many other PTMs that septins are subject to (phosphorylation, acetylation, etc.). Their possible involvement in septin ring splitting or other septin-related processes is an important subject for future studies.

5. Fig. 2: Hof1 protein levels across cell cycle stages for the different Hof1 alleles should be shown for a comparison alongside the cell-cycle dependent phosphorylation events. Also, Hof1 has previously shown to be degraded by Grr1 during late mitosis and G1 phase through its PEST motif (Blondel et. al, 2005). Since some of the phosphorylation sites studied here lie in or near the PEST motif, the authors should comment on its significance with respect to their observations.

We have now carried out a careful analysis of Hof1-3HA protein levels and phosphorylation throughout the cell cycle in wild type *HOF1* cells, as well as in the *HOF1-P1A*, *HOF1-P1E*, *HOF1-P4A* and *HOF1-P4E* mutants. The data are shown in the new Fig. S4A-B and show that Hof1 protein levels are not dramatically affected in any cell cycle phase in the mutants. Phosphorylation is only mildly reduced in *HOF1-P1A* and *HOF1-P4A* relative to wild type cells (Fig. S4A-B; compare the ratio between phosphorylated and unphosphorylated isoforms in wild type versus *HOF1-P1A*/*-P4A* cells at 90-105'). In contrast, the *HOF1-P1E* and *HOF1-P4E* mutant alleles cause a constitutive electrophoretic shift in the corresponding proteins that was already noticed in other experiments (see Fig. 2C, 3A, 4B, S5A-B). In spite of this, these mutated proteins still undergo cell cycle-dependent phosphorylation, which is consistent with the notion that multiple protein kinases phosphorylate Hof1.

6. Fig.3: The localization of Hof1 proteins (WT, 4A, 4E) has not been shown when *Dma2* is overexpressed. Does it change as compared to what is shown in Fig.3C?

This was an important point. We have now analysed the localization of wild type Hof1-eGFP in cells overexpressing *DMA2* with the same experimental set-up that we used to analyse its distribution in wild type cells (see point #1; new Fig. 1C). Surprisingly, instead of localizing on the septin collar as a double ring during mitosis, the Hof1-eGFP signal colocalised homogeneously with the whole septin collar throughout mitosis, thus precluding the assessment of its possible translocation to the AMR. We think that this may be due to the increased Hof1 protein levels in *DMA2*-overexpressing cells (see point #2; Fig. S7A), because a similar distribution was previously found using a Hof1 mutant lacking its PEST motif (Blondel et al., 2005). The data on Hof1 localisation upon *DMA2* overexpression are shown in the new Fig. S7B. Due to this unexpected and inconclusive result, we did not proceed to analyse the localization of Hof1-P4E in *GAL1-DMA2m* cells.

7. Fig. 4: Fig.4B suggests that deletion of the F-BAR domain is important for Hof1 degradation and causes an increase in Hof1 levels. Could the increased Hof1 levels be the reason why Hof1- Δ F-BAR phospho-mimetic alleles do not show a rescue of phenotype as represented in Fig. 4D.

Establishing if deletion of the F-BAR domain impairs Hof1 degradation would require additional experiments. We are cautious about this conclusion because the Hof1(Δ F-BAR) protein is considerably smaller in size and could therefore get better transferred to nitrocellulose in western blotting. However, we think unlikely that Hof1 stabilisation may affect its ability to promote septin ring splitting for several reasons. First, we show that the Hof1(Δ F-BAR) protein translocates to the AMR with normal kinetics and levels, suggesting that it dissociates from septins at the right time. Second, Hof1 reaches its maximal levels at the bud neck around the time of septin splitting, arguing against a role of protein degradation in Hof1 relocation from the septin collar to the AMR. Consistently, Hof1 degradation was found not to be required for septin ring splitting (Blondel et al., 2005). Third, our data indicate that the bulk of Hof1 degradation likely occurs at the M/G1 transition, which is later than septin splitting. Fourth, we have now measured the stability of Hof1, wild type and P4A/P4E mutants, by cycloheximide chase experiments. Our data indicate that both the -P4A and -P4E mutations both lead to Hof1 stabilisation in spite of their different phenotypic effects *in vivo*. These data are shown in the new Fig. S5B.

8. Fig. 5: The authors have performed *in-vitro* analysis of Hof1 effect on septin filaments reconstituted from purified septins. It would be more informative to add how the Hof1-P4A, Δ F-BAR and Hof1- Δ FBAR-phosphomimetic(E) mutant protein affect septin filament bundling. Hof1- Δ SID (septin-interacting domain) should also tested in this experiment.

We have carried out a considerable amount of work to answer this point. Our data show that Hof1-P4A has septin-bundling activity similar to that of wild type Hof1 (new Fig. S8B), in agreement with the presumed lack of phosphorylation of recombinant Hof1. In contrast, Hof1(Δ SID) and Hof1(Δ F-BAR) are severely impaired in their ability to bundle septins (new Fig. S8B), which is likely linked to their reduced affinity for septins. Indeed, not only the SID but also the F-BAR domain was previously shown to contribute to septin binding (Meitinger et al., 2011). Since both Hof1-4PE and Hof1(Δ F-BAR) are defective in septin bundling, we did not test a Hof1(Δ F-BAR)-P4E protein in our *in vitro* assays.

9. In lines 292-296, the authors hypothesize that F-BAR domain of Hof1 may modify the plasma membrane at the AMR after translocation. But the question arises that the F-BAR domain is also present in the Hof1 WT and P4A proteins and these variants also show a translocation to the medial AMR, then why are these variants of Hof1 unable to rescue the no-splitting defect? This should be discussed and clarified.

This is a good point for which we do not have a definitive answer, since we could not establish if and how Hof1 localisation is perturbed upon *DMA2* overexpression. We speculate that in these conditions Hof1 phosphorylation may be impaired causing its retention on the septin collar. Unfortunately, a number of experimental caveats did not allow us to formally prove this idea (see our answers to points #2 and #6).

Minor comments:

1. Fig1-B: Does the chain phenotype exhibiting no septin splitting represent the population of cell with the cell cycle arrest or the delay in cell cycle progression due to inactivated MEN signalling?

We previously characterized the phenotype of *GAL1-DMA2m* cells (Fraschini et al., 2004) and found that, at least during the first cell cycle after galactose induction, cells progress through and exit from mitosis with kinetics similar to wild type cells. In spite of this, the septin collar is not rearranged and cells enter into a new cell cycle, forming a new bud and a new septin collar and experiencing cytokinesis defects. Most recently (Tamborrini et al., 2018), we have shown that Dma2 ubiquitylates the MEN scaffold at spindle pole bodies, Nud1, and hampers the recruitment of MEN proteins to this location, thereby interfering with MEN signaling to levels that do not cause mitotic exit problems but only inhibit septin ring splitting and cytokinesis. Very likely, different thresholds of MEN signaling and Cdc14 activation are needed for mitotic exit versus cytokinesis.

2. Authors should represent the Fig1-E data prior to Fig-D. And both the panels should indicate whether the inducer is IAA or NAA for Hof1-3xminiAID.

The reason why we put the two panels in this order is because we first tried to deplete Hof1-3miniAID with 1mM IAA (Fig. 1D), which is a concentration that is often used in publications and leads to efficient depletion. However, this concentration is highly toxic under the laser illumination regimes that we used for our live cell imaging. This is why we switched to 0.1mM NAA, which is far less toxic for yeast cells during imaging (Papagiannakis et al., 2017), and tested if these conditions would efficiently deplete Hof1-3xminiAID (Fig. 1E). We

have now added the inhibitor in the Fig. 1F-G panels.

3. Authors have not specified the type of inducer (IAA or NAA) used in Fig1-F and Fig1-G or in the text. Authors should represent the data with NAA in all these panels.

Thank you for noticing this. We have now modified the text and the figure accordingly.

Reviewer #1 (Remarks to the Author):

I am satisfied with the revisions made to the manuscript and the responses to my concerns and comments. This study represents a tremendous amount of work and I commend the authors for their thorough attention to detail.

Reviewer Identity: Michael McMurray, PhD, University of Colorado Anschutz Medical Campus

Reviewer #2 (Remarks to the Author):

The authors have done a great job of addressing the concerns expressed by all of the reviewers. Most importantly, they have added quantitation and statistical tests to many of the figures where it was missing, and performed a number of new experiments that addressed concerns. The revised manuscript is much stronger, and should be published.

Reviewer #3 (Remarks to the Author):

The authors have very thoroughly responded to the reviewers' comments. They are well-documented, with additional new experiments, and the revised manuscript has my endorsement for publication in Nature Communications.